# Synthesis and Significance of Arachidonic Acid, a Substrate for Cyclooxygenases, Lipoxygenases, and Cytochrome P450 Pathways in the Tumorigenesis of Glioblastoma Multiforme, Including a Pan-Cancer Comparative Analysis

**DOI:** 10.3390/cancers15030946

**Published:** 2023-02-02

**Authors:** Jan Korbecki, Ewa Rębacz-Maron, Patrycja Kupnicka, Dariusz Chlubek, Irena Baranowska-Bosiacka

**Affiliations:** 1Department of Biochemistry and Medical Chemistry, Pomeranian Medical University in Szczecin, Powstańców Wlkp. 72, 70-111 Szczecin, Poland; 2Department of Ecology and Anthropology, Institute of Biology, University of Szczecin, Wąska 13, 71-415 Szczecin, Poland

**Keywords:** glioblastoma multiforme, arachidonic acid, fatty acid, PUFA, prostaglandin, leukotriene, 5-HETE, cyclooxygenase-2, 5-lipoxygenase

## Abstract

**Simple Summary:**

Glioblastoma multiforme is a brain tumor with a very unfavorable prognosis, where the vast majority of patients do not survive a year after diagnosis. One line of research that may help in designing more successful therapeutic approaches is the synthesis and metabolism of arachidonic acid, which is then converted into a large number of different lipid mediators, including prostaglandins and leukotrienes (by cyclooxygenases and lipoxygenases, respectively). In this paper, we discuss the synthesis of arachidonic acid in glioblastoma multiforme tumors as well as the significance of lipid mediators synthesized from arachidonic acid, which can increase the proliferation of glioblastoma multiforme cancer cells, cause angiogenesis, inhibit the anti-tumor response of the immune system, and be responsible for resistance to treatment.

**Abstract:**

Glioblastoma multiforme (GBM) is one of the most aggressive gliomas. New and more effective therapeutic approaches are being sought based on studies of the various mechanisms of GBM tumorigenesis, including the synthesis and metabolism of arachidonic acid (ARA), an omega-6 polyunsaturated fatty acid (PUFA). PubMed, GEPIA, and the transcriptomics analysis carried out by Seifert et al. were used in writing this paper. In this paper, we discuss in detail the biosynthesis of this acid in GBM tumors, with a special focus on certain enzymes: fatty acid desaturase (FADS)1, FADS2, and elongation of long-chain fatty acids family member 5 (ELOVL5). We also discuss ARA metabolism, particularly its release from cell membrane phospholipids by phospholipase A_2_ (cPLA_2_, iPLA_2_, and sPLA_2_) and its processing by cyclooxygenases (COX-1 and COX-2), lipoxygenases (5-LOX, 12-LOX, 15-LOX-1, and 15-LOX-2), and cytochrome P450. Next, we discuss the significance of lipid mediators synthesized from ARA in GBM cancer processes, including prostaglandins (PGE_2_, PGD_2_, and 15-deoxy-Δ^12,14^-PGJ_2_ (15d-PGJ_2_)), thromboxane A_2_ (TxA_2_), oxo-eicosatetraenoic acids, leukotrienes (LTB_4_, LTC_4_, LTD_4_, and LTE_4_), lipoxins, and many others. These lipid mediators can increase the proliferation of GBM cancer cells, cause angiogenesis, inhibit the anti-tumor response of the immune system, and be responsible for resistance to treatment.

## 1. Introduction

Glioblastoma multiforme (GBM) is one of the most aggressive brain tumors and has the worst prognosis, with an average survival of about one year [1,2,3]. In order to either improve existing therapies or develop new approaches, the mechanisms of GBM tumorigenesis are being intensively investigated, including those involving arachidonic acid (ARA) C20:4n-6 and the lipid mediators formed from this fatty acid.

PUFAs, in particular arachidonic acid ARA C20:4n-6, eicosapentaenoic acid (EPA) C20:5n-3, and docosahexaenoic acid (DHA) C22:6n-3, can be converted into lipid mediators, such as eicosanoids [4], and pro-resolving lipid mediators [5]. Eicosanoids are 20-carbon lipid mediators synthesized from ARA C20:4n-6, dihomo-γ-linolenic acid C20:3n-6, and EPA C20:5n-3 using cyclooxygenases (COX) and lipoxygenases (LOX), resulting in the formation of prostaglandins and leukotrienes, respectively [4]. Eicosanoids have pro-inflammatory properties, although there are also lipid mediators with anti-inflammatory properties, such as 15-deoxy-Δ^12,14^-PGJ_2_ (15d-PGJ_2_) [6]. EPA and DHA can be converted into pro-resolving lipid mediators with LOX, cytochrome P450, and acetylated cyclooxygenase-2 (COX-2) [5]. This conversion produces lipoxins and resolvins, although it should be mentioned here that free PUFAs, including ARA, are the activators of peroxisome proliferator-activated receptors (PPAR)α and PPARγ [7].

All of the aforementioned groups of ARA metabolites have either pro- or anti-cancer properties in GBM tumors, which indicates their significance in GBM tumor development. Despite their important role, some groups of these lipid mediators are little-known and rarely studied, and there is no paper in the literature that reviews the body of research in this area. The aim of this paper is to fill this gap and at the same time generate more interest in the role of ARA metabolites in GBM.

## 2. Methodology

This study’s major objective is to characterize the significance of all ARA C20:4n-6-derived lipid mediators, their receptors, and the enzymes responsible for their production in the tumorigenic pathways in GBM. The PubMed search engine (https://pubmed.ncbi.nlm.nih.gov accessed on 1 October 2022) was used for this purpose. Due to the fact that many of the lipid mediators produced from ARA C20:4n-6 have not yet been investigated in the context of GBM, two additional sources were used to conduct a bioinformatic analysis of every gene in GBM, namely, the transcriptomics analysis carried out by Seifert et al. [8] and the Gene Expression Profiling Interactive Analysis (GEPIA) web server (http://gepia.cancer-pku.cn accessed on 20 October 2022) [9].

The analyses posted on the GEPIA portal include the analysis of nearly 10,000 samples from 33 different cancers deposited in the Cancer Genome Atlas (TCGA) [10] along with the analysis of more than 8000 healthy tissue samples posted in Genotype-Tissue Expression (GTEx) [11,12]. The GEPIA served as a source of data on differences in the expression of given genes between GBM tumor and healthy brain tissue, and for linking the expression of a given gene to GBM patient prognosis.

A transcriptomics analysis was performed by Seifert et al. [8] on nearly 17,000 different genes in various grades of glioma, including GBM, from 45 patients. These results were normalized with a control: an analysis of gene expression in brain samples from 21 epilepsy patients from the REpository of Molecular BRAin Neoplasia DaTa (Rembrandt) [13], which served as a second source of data on differences in the expression of genes between GBM tumors and healthy brain tissue.

## 3. Arachidonic Acid Biosynthesis and Glioblastoma Multiforme

### 3.1. Arachidonic Acid Biosynthesis

ARA C20:4n-6 in humans is not synthesized de novo but from linoleic acid C18:2n-6 in the PUFA biosynthesis pathway (Figure 1) [14]. Linoleic acid C18:2n-6 in its activated form, linoleoyl-CoA C18:2n-6, undergoes desaturation with fatty acid desaturase 2 (FADS2)/Δ^6^-desaturase (D6D), which is accompanied by the formation of γ-linolenoyl-CoA C18:3n-6. Subsequently, the hydrocarbon chain in this fatty acyl-CoA is elongated with two carbons through the elongation of the long-chain fatty acid family members 5 (ELOVL5), accompanied by the formation of dihomo-γ-linolenoyl-CoA C20:3n-6. At the same time, an alternative pathway for the synthesis of dihomo-γ-linolenoyl-CoA C20:3n-6 from linoleoyl-CoA C18:2n-6 is also possible [15]. Linoleoyl-CoA C18:2n-6 is first elongated with ELOVL5 and then desaturated by FADS2. This means that these two enzymes can catalyze the formation of dihomo-γ-linolenoyl-CoA C20:3n-6 in reverse order. In this alternative pathway of PUFA biosynthesis, FADS2 shows activity not of Δ^6^-desaturase but of Δ^8^-desaturase. In the latter reaction, the hydrocarbon chain in dihomo-γ-linolenoyl-CoA C20:3n-6 is desaturated with fatty acid desaturase 1 (FADS1)/Δ^5^-desaturase (D5D), which is accompanied by the production of arachidonyl-CoA C20:4n-6. In the same way as arachidonyl-CoA C20:4n-6, EPA-CoA C20:5n-3 can also be synthesized from α-linolenoyl-CoA C18:3n-3 [14]. Arachidonyl-CoA C20:4n-6 is an activated form of ARA that participates in metabolic pathways, including lipid synthesis pathways. Once synthesized, arachidonyl-CoA C20:4n-6 is used to make lipids, particularly phospholipids. Incorporated into phospholipids, ARA C20:4n-6 is stored and then released by phospholipases A_2_ (PLA_2_) as a free fatty acid [16]. Arachidonyl-CoA C20:4n-6 can also be further elongated via elongation of the long-chain fatty acid family members 2 (ELOVL2) and ELOVL5 in a synthesis pathway similar to the synthesis of DHA C22:6n-3 from EPA C20:5n-3 [14,17,18,19].

### 3.2. Arachidonic Acid Biosynthesis Pathway in Glioblastoma Multiforme Tumors

Expression of FADS2, an enzyme important for the viability and self-renewal of GBM cancer stem cells [20], is higher in GBM tumors than in healthy brain tissue, according to GEPIA [9] and the transcriptomics analysis performed by Seifert et al. [8]. However, our study showed that FADS2 may have lower expression in tumors than in the peritumoral area in GBM patients [21]. Discrepancies between our results and the data from GEPIA and transcriptomics analysis performed by Seifert et al. may have resulted from studying different groups of patients. FADS2 expression in GBM tumors does not differ between men and women [21]. According to the GEPIA portal, higher FADS2 expression does not affect the prognosis for GBM patients [9]. Studies in GBM models show that FADS2 expression is higher in GBM cancer stem cells than in other GBM cancer cells [20].

The expression of FADS1, which is also important for the viability and self-renewal of GBM cancer stem cells [20], does not differ between GBM tumors and healthy brain tissue, according to GEPIA [9], Seifert et al. [8], and previous results from our research team [21]. According to the GEPIA portal, a higher FADS1 expression does not affect the prognosis for GBM patients [9]. FADS1 expression is higher in GBM cancer stem cells than in other GBM cancer cells [20].

ELOVL5 expression is higher in GBM tumors compared to healthy brain tissue, according to GEPIA [9] and Seifert et al. [8]. However, previous results from our research team did not show significant differences in the expression of ELOVL5 in GBM tumor tissue versus the peritumoral area [22]. Discrepancies between our results and the data from GEPIA and transcriptomics analysis performed by Seifert et al. may have resulted from studying different groups of patients. In addition, we observed that ELOVL5 expression was lower in GBM tumors in women relative to both the peritumoral area and GBM tumors in men [22]. Higher ELOVL5 expression does not affect the prognosis for GBM patients, according to GEPIA [9]. ELOVL5 expression can be higher in a GBM tumor as a result of hypoxia, as shown by our experiments with U87 MG line cells [22]. This is very important because hypoxia in a GBM tumor also increases the expression of COX-2 [23], an enzyme that converts ARA into prostanoids. This means that hypoxia increases the production of ARA and, at the same time, its conversion into prostanoids.

## 4. Phospholipase A_2_ Superfamily and the Release of Arachidonic Acid from Cell Membrane Phospholipids in Glioblastoma Multiforme

### 4.1. Phospholipase A_2_ Superfamily

The production of prostaglandins and leukotrienes requires a substrate for COX and LOX, namely, free ARA C20:4n-6, which is cleaved from cell membrane phospholipids by PLA_2_. Enzymes with PLA_2_ activity cleave either a fatty acid or a short acyl group from phospholipids at the *sn*-2 position [16]. All of these enzymes form the phospholipase A_2_ superfamily, which can be divided into six types. Three of these types are important in the release of ARA C20:4n-6 as well as other PUFA from cell membrane phospholipids [16]:cytosolic phospholipase A_2_ (cPLA_2_),calcium-independent phospholipase A_2_ (iPLA_2_), andsecretory phospholipase A_2_ (sPLA_2_).

The remaining PLA_2_ types include:platelet-activating factor acetyl hydrolases (PAF-AH),lysosomal phospholipase A_2_, andadipose phospholipase A_2_.

In humans, seven representatives of cPLA_2_ are distinguished, namely, cPLA_2_α/*PLA2G4A* to cPLA_2_ζ/*PLA2G4F*. These enzymes, activated by Ca^2+^ [16], belong to the group IV (GIV) PLA_2_. Significantly, cPLA_2_γ/*PLA2G4C* lacks a Ca^2+^ binding domain and is not sensitive to this second messenger [24]. cPLA_2_α is additionally activated by phosphorylation and has the highest activity towards phosphatidylcholine (PC), phosphatidylethanolamine (PE), and, to a lesser extent, towards other glycerophospholipids [16]. cPLA_2_ have a specificity for cleaving PUFA from glycerophospholipids, particularly ARA C20:4n-6. cPLA_2_α shows the highest specificity for cleaving ARA C20:4n-6 [25,26], to a lesser extent, EPA C20:5n-3, and, to an even lesser extent, other PUFAs, e.g., linoleic acid C18:2n-3. cPLA2γ also has the highest specificity for cleaving ARA C20:4n-6 and a twice-lower specificity for cleaving both linoleic acid C18:2n-3 and oleic acid C16:1n-9 [26].

In humans, there are six representatives of iPLA_2_: iPLA_2_β to iPLA_2_η [16]. All of these enzymes belong to the GVI PLA_2_. They are activated by ATP [27], and their activity is independent of Ca^2+^ levels and reduced by calmodulin [28]. Enzymes in this group show different specificities for cleaving fatty acids from phospholipids at the *sn*-2 position. Depending on the enzymes, they show a higher ability to release a given fatty acid, e.g., oleic acid C16:1n-9 [27] or ARA C20:4n-6 [29].

Seventeen different groups of PLA_2_ have been classified to date, which includes sPLA_2_ [16]. Some sPLA_2_ groups consist of only the sPLA_2_ found in the venom of snakes, insects such as bees, and scorpions [16,30,31,32]. In humans, there are nine representatives of sPLA_2_ [16]. These enzymes cleave fatty acids from phospholipids at the *sn*-2 position without showing specificity to a particular fatty acid [16,33]. Once secreted into the intercellular space, sPLA_2_ not only cause the release of ARA C20:4n-6 but can also activate their receptor PLA_2_R1 [34].

After fatty acids are cleaved from phospholipids by PLA_2_, free fatty acids are formed, most commonly ARA C20:4n-6 and lysophosphatidylcholine (LPC) if PC was the reaction substrate (Figure 2). LPC can then be converted to lysophosphatidic acid (LPA) by the action of enzymes with lysophospholipase D (lysoPLD) activity [35,36]. An extracellular enzyme with lysoPLD activity is autotaxin (ATX)/ENPP2 [35,36]. Importantly, if the substrate for PLA_2_ is phosphatidic acid (PA), then LPA is formed directly [37]. LPA is a lipid mediator that acts through its six receptors (from lysophosphatidic acid receptor 1 (LPAR_1_) to LPAR_6_) [38].

### 4.2. Cytosolic Phospholipase A_2_ and Calcium-Independent Phospholipase A_2_ in Glioblastoma Multiforme

Expression of cPLA_2_α/*PLA2G4A* is upregulated in GBM tumors compared to healthy brain tissue [39]. This is also confirmed by bioinformatics analysis on the GEPIA portal [9] and the transcriptomics analysis by Seifert et al. [8]. At the same time, the expression of cPLA_2_β/*PLA2G4B* is lower, and the expressions of cPLA_2_γ/*PLA2G4C*, cPLA_2_δ/*PLA2G4D*, cPLA_2_ε/*PLA2G4E*, and cPLA_2_ζ/*PLA2G4F* are unchanged, according to GEPIA [9]. The expression of cPLA_2_γ/*PLA2G4C* is lower, and cPLA_2_ζ/*PLA2G4F* is not different in GBM tumors relative to healthy brain tissue, according to the transcriptomics analysis by Seifert et al. [8]. For six of the iPLA_2_, expression in GBM tumor does not differ compared to healthy brain tissue, according to GEPIA [9]. The expression of iPLA_2_β/*PLA2G6* and iPLA_2_δ/*PNPLA6* is lower in GBM tumor than in the healthy brain, according to the transcriptomics analysis by Seifert et al. [8]. Expressions of the remaining iPLA_2_ do not differ between GBM tumors and healthy brain tissue.

In the case of iPLA_2_η/*PNPLA4*, higher expression in GBM tumors is associated with a worse prognosis for the patient, according to GEPIA (Table 1) [9]. For iPLA_2_ζ/*PNPLA2*, there is a trend (*p* = 0.087) of worse prognosis and higher expression of this gene in the GBM tumor.

cPLA_2_ are activated in GBM cells, in particular, by sPLA_2_ enzymes [40,41]. This is associated with the induction of cPLA_2_ phosphorylation via MAPK kinase cascades as well as with an increase in cytoplasmic Ca^2+^ levels via phospholipase C-γ (PLC-γ) activation.

cPLA_2_α increases the proliferation of GBM cells, although the effect is not large. The most significant property of cPLA_2_α in GBM cells is causing chemoresistance to temozolomide (TMZ) and other chemotherapeutics, such as doxorubicin and 5-fluorouracil [39]. At the same time, the increased activity of cPLA_2_ may also decrease the viability of GBM cells, where TMZ induces the phosphorylation of cPLA_2_. This increases the activation of this enzyme [42] and thus leads to an increase in the level of free ARA 20:4n-6, whose excess reduces the viability of GBM cells. The reason for this may be in the activation of PPAR by this fatty acid [7,43,44] and the generation of reactive oxygen species (ROS) [45].

PLA_2_ may also be important in the interaction of GBM cells with endothelial cells. GBM cells cause an increase in the expression and activity of cPLA_2_ and iPLA_2_ in endothelial cells [46,47]. An increase in cPLA_2_ activity in endothelial cells can also be caused by radiation therapy [48]. A rise in the activity of cPLA_2_ and iPLA_2_ leads to the production of LPA [49]. GBM cancer cells may also increase COX-2 expression in endothelial cells, which increases the production of prostanoids including prostaglandin E_2_ (PGE_2_) [47]. LPA and PGE_2_ increase the proliferation and migration of endothelial cells [46,47,49]. This is also a mechanism of angiogenesis as a side effect of GBM radiotherapy [47,48]. At the same time, angiogenesis can be inhibited by pericytes [47].

Dying endothelial cells in a GBM tumor can secrete PGE_2_ that increases the proliferation of GBM cells [50]. This is associated with the processing of iPLA_2_β by caspase 3 [16,51], which increases the activity of this iPLA_2_ and, thus, leads to an increase in PGE_2_ production [50].

### 4.3. Secretory Phospholipase A_2_ in Glioblastoma Multiforme

Analyses on the GEPIA portal indicate that PLA2G5 expression is higher in GBM tumors [9]. There is also elevated expression of *PLA2G2A*, *PLA2G12A*, and *PLA2G15* but no other sPLA_2_ in GBM tumors [9]. The transcriptomics analysis by Seifert et al. showed that the expressions of *PLA2G2A* and PLA2G5 are higher in GBM tumors than in healthy brain tissue [8]. This is the same as the data from the GEPIA web server. However, Seifert et al. showed that the expression of *PLA2G12A* and of the other sPLA_2_ enzymes is not different in GBM tumors relative to healthy brain tissue [8]. Wu et al. also showed that *PLA2G5* expression is higher in gliomas than in healthy tissue and increases with tumor grade [52].

Higher expression of certain sPLA_2_ in GBM tumors is associated with a worse prognosis. According to GEPIA, these include *PLA2G1B* and *PLA2G15* [9]. Wu et al. showed a higher number of sPLA_2_ affecting prognosis. In particular, worse prognoses in patients with GBM are associated with higher expression of *PLA2G1B*, *PLA2G2E*, *PLA2G3*, and *PLA2G5* [52].

*PLA2G5* is significant for tumorigenesis in low-grade gliomas and GBM. This suggests that a high expression of this sPLA_2_ is associated with a worse prognosis in patients with GBM and low-grade gliomas (Table 2) [52]. Analyses on the GEPIA portal show no significant association between the expression of the aforementioned sPLA_2_ and the GBM patient prognosis [9].

sPLA_2_ are secreted outside the cells where they perform their function. They have their own receptor, PLA_2_R1, from the C-type lectin superfamily and mannose receptor family [34], located in the cell membrane, through which it passes once. According to both GEPIA [9] and Seifert et al. [8], PLA_2_R1 expression does not differ between GBM tumors and healthy brain tissue. An above-average expression of this receptor in a GBM tumor is associated with a worse prognosis for the patient [9], indicating that sPLA_2_ may act on PLA_2_R1 and be pro-tumorigenic.

sPLA_2_ may act by participating in the production of LPA, a lipid mediator that has six different receptors [38]. According to GEPIA, LPAR_3_ expression is downregulated in GBM tumors relative to healthy brain tissue [9], whereas LPAR_5_ and LPAR_6_ expression is upregulated in GBM tumors. The expression of other LPA receptors is not altered in GBM tumors. The transcriptomics analysis by Seifert et al. shows that LPAR_1_ expression is lower, and LPAR_6_ expression is higher in GBM tumors relative to healthy brain tissue [8]. The expression of other LPA receptors does not differ between GBM tumors and healthy brain tissue.

sPLA_2_ also have the same catalytic properties as other PLA_2_. They cause the release of ARA 20:4n-6 from cell membrane phospholipids; this reaction produces free ARA 20:4n-6 and LPC. The latter is converted into LPA in the intercellular space by ATX [53], which is secreted by GBM cancer cells [54,55] and whose expression in GBM tumors is higher than in healthy brain tissue [53] and is elevated by interaction with microglial cells [55]. At the same time, GEPIA reports that ATX expression is not altered in GBM tumors [9], and Seifert et al. showed that it is lower [8] than in healthy brain tissue. The level of ATX expression in the tumor is not associated with prognosis severity for patients with GBM [9].

Another important source of ATX in the GBM tumor microenvironment is microglial cells [55], where ATX expression is upregulated by GBM cells, especially under hypoxia. Microglial cells also express the LPAR_1_ receptor and can respond to LPA [55].

Increased expression of various sPLA_2_ [52] and ATX [53] in GBM tumors also results in increased LPA production. GBM cancer cells show a loss of primary cilia, which leads to an increase in the distribution of LPAR_1_ in the plasma membrane of these cells and to an enhancement of signal transduction by this receptor as a result of a greater association of G proteins with this receptor [56].

LPA causes GBM cells to migrate [53,54,55,57,58] due to the activation of LPAR_1_, which results in the activation of protein kinase C (PKC)α. This is responsible for the phosphorylation of the progesterone receptor at the Ser^400^ residue [59,60]. GBM cancer cell migration is also facilitated by the LPA-induced decrease in oligodendrocyte adhesion [54]. It is also worth mentioning that in addition to LPAR_1_, the receptor for advanced glycation end products (RAGE) may be another important receptor causing GBM cancer cell migration [61].

LPA increases the proliferation of GBM cancer cells [55]. The effect of LPA on proliferation depends on LPAR_1_ receptors [55] and RAGE [61], and it occurs via the activation of two signaling pathways. The first is the Rho → sodium-hydrogen antiporter 1 (NHE-1) pathway, which leads to an increase in intracellular pH and, thus, the proliferation of GBM cancer cells [62]. The second pathway is the activation of extracellular signal-regulated kinase (ERK) mitogen-activated protein kinase (MAPK) by the phosphatidylinositol-4,5-bisphosphate 3-kinase (PI3K) → PKC pathway [62], which can also be initiated by epidermal growth factor receptor (EGFR) transactivation. Studies on PLA_2_G2A have shown that this sPLA_2_ increases GBM cancer cell proliferation via EGFR transactivation [63,64,65]. This is associated with the activation of PKC, which activates EGFR [64]. EGFR activation results in the activation of the Src → ERK MAPK → Akt/PKB → mammalian target of rapamycin (mTOR) → ribosomal protein 70 S6 kinase (p70S6K) pathway [63,65]. Its consequence is an increase in the proliferation of GBM cancer cells.

sPLA_2_ can also increase GBM cancer cell proliferation indirectly through the activation of cPLA_2_ inside a GBM cell [40]. This process is independent of LPA.

LPA inhibits FasL-induced apoptosis [66] due to the LPA-induced activation of thyroid hormone receptor-interacting protein 6 (TRIP6). TPIP6 binds directly to Fas receptor (FasR)/CD95, which inhibits the induction of apoptosis by this receptor [66].

LPA causes radioresistance of GBM cancer cells [48,58]. These effects are a result of the LPA-induced activation of LPAR_1_ [53,55] and LPAR_3_ [48].

Phosphorylation of the progesterone receptor by LPA increases vascular endothelial growth factor (VEGF) expression in GBM cancer cells [60], the most important growth factor in angiogenesis. LPA is also important in radiotherapy-induced angiogenesis in GBM tumors [58]. An increase in tumor vascularization during exposure to ionizing radiation can be inhibited by ATX inhibitors, which could have some clinical application in future therapies against GBM [58].

The aforementioned actions of LPA were carried out on various models of specific GBM cell lines. Significantly, the action of LPA may be more pronounced in GBM cancer stem cells than non-cancer stem cells, as the former show much higher expression of LPAR_1_ and LPAR_3_ [67].

LPAR_1_ is important in the development of GBM. Higher expression of this receptor in GBM tumors is associated with a worse prognosis [55]. At the same time, an analysis on the GEPIA portal did not link LPAR_1_ and LPAR_3_ expression to prognosis severity for GBM patients [9]. In addition, it did not show that the expression of the other LPA receptors had an effect on the prognosis for GBM patients.

### 4.4. Pan-Cancer Analysis of Phospholipase A_2_ Genes and Comparison of GBM Expression against Other Cancers

We also performed a pan-cancer analysis of the expression of the PLA_2_ genes with the GEPIA portal [9].

In GBM, but not in lower grade gliomas, there is higher expression of cPLA_2_α/*PLA2G4A* compared to healthy brain tissue [8,9]. Among the analyzed 31 tumor types, only four more had higher expression of this PLA_2_, and eight other types showed a decrease. For this reason, higher expression of this enzyme in GBM tumors can be considered characteristic for this cancer.

In GBM, the expression of cPLA_2_ β/*PLA2G4B* is decreased relative to healthy brain tissue [9], similar to lower grade glioma and 19 other types of cancer. This indicates that the decreased expression of this PLA_2_ in tumor is a hallmark of cancer.

Seifert et al. also indicates that cPLA_2_γ/*PLA2G4C* expression may be downregulated in GBM tumors relative to healthy brain tissue [8]. According to a pan-cancer analysis based on the GEPIA, cPLA_2_γ/*PLA2G4C* expression is downregulated in nine types of tumors but not in GBM or lower grade gliomas, whereas it is upregulated in seven types of tumors [9]. Changes in cPLA_2_γ/*PLA2G4C* expression in GBM tumors could be a hallmark of cancer.

Seifert et al. also showed a decrease in the expression of iPLA_2_β/*PLA2G6* and iPLA_2_δ/*PNPLA6* in GBM tumors relative to healthy brain tissue [8]. According to GEPIA, iPLA_2_β/*PLA2G6* expression is downregulated in 15 tumor types (Table 3) [9], whereas iPLA_2_δ/*PNPLA6* expression is only downregulated in three types. For this reason, it can be thought that decreased iPLA_2_β/*PLA2G6* expression may be a hallmark of cancer. In contrast, reduced expression of iPLA_2_δ/*PNPLA6* is characteristic of GBM.

Available sources [8,9] show that *PLA2G2A*, *PLA2G5*, *PLA2G12A,* and *PLA2G15* undergo increased expression in GBM relative to healthy brain tissue. Changes in sPLA_2_’s expression in GBM are characteristic of this cancer. All listed sPLA_2_ undergo increased expression only in certain types of cancer (apart from GBM): *PLA2G2A* (in 2); *PLA2G5* (in 1); *PLA2G12A* (in 4); *PLA2G15* (in 3).

*PLA2G2A* expression is downregulated in 18 out of 31 types of cancer, indicating that it is generally downregulated in cancer (Table 4). In contrast, increased expression of *PLA2G2A* may occur in GBM [8,9], which may be characteristic of GBM. On the other hand, in 17 out of 31 cancers, there is a higher expression of *PLA2G7* in the tumor than in healthy tissue. Its expression in a GBM tumor is not different from its expression in healthy brain tissue [8,9].

### 4.5. Lysophospholipid Acyltransferases in Glioblastoma Multiforme

When discussing the importance of PLA_2_ in tumorigenesis in GBM, it is also important to mention enzymes that catalyze the opposite reaction to the enzymes in question. An example of this is lysophosphatidylcholine acyltransferases (LPCAT), which catalyze the opposite reaction towards PC [68]. LPCAT causes the formation of PC from LPC and fatty acyl-CoA. For this reason, LPCAT decreases the level of LPA, a lipid mediator important in cancer processes in GBM. According to the GEPIA portal, GBM tumors have higher expressions of LPCAT1, LPCAT2, and LPCAT3, but lower expression of LPCAT4/LPEAT2 relative to healthy brain tissue [9]. In addition, according to Seifert et al., the expression of LPCAT1 and LPCAT3 is higher in GBM tumors than in healthy brain tissue [8]. In contrast, LPCAT4 expression is lower in GBM tumors. This confirms the results obtained from the GEPIA database. An increase in the expression of the aforementioned enzymes may contribute to a decrease in LPA level but also contribute to the intense remodeling of phospholipids in the cell membranes of GBM cells. At the same time, according to the GEPIA database, the expression of the mentioned enzymes does not affect the prognosis severity of GBM patients [9].

### 4.6. Acyl-CoA Thioesterases and Arachidonic Acid C20:4n-6 in Glioblastoma Multiforme

The most important pathway for the formation of free ARA C20:4n-6 is through PLA_2_ activity. However, free ARA C20:4n-6 can be formed from hydrolysis of arachidonyl-CoA by acyl-CoA thioesterases (ACOT) [69], a group of nine enzymes that cause hydrolysis of fatty acyl-CoA to free fatty acid and CoA [69,70]. An example of an enzyme from this group is ACOT7, which shows activity towards arachidonyl-CoA and saturated fatty acyl-CoA [69,70,71]. According to GEPIA and Seifert et al., there is a reduction in ACOT7 expression in GBM tumors relative to healthy brain tissue [8,9], where higher expression of this enzyme is associated with a worse prognosis for a GBM patient [9], suggesting the involvement of ACOT7 in tumorigenesis in GBM.

According to GEPIA and Seifert et al., there is also elevated expression of ACOT9 in GBM tumors [8,9], an enzyme showing the highest activity to myristoyl-CoA [69,70,72] and low activity to longer acyl-CoA. Importantly, the expression level of ACOT9 is not associated with the prognosis for a patient with GBM [9]. According to GEPIA, the expression of other ACOT does not differ between GBM tumors and healthy brain tissue [9]. In addition, Seifert et al. indicate that the expression of ACOT4 and ACOT8 in GBM tumors is lower than in healthy brain tissue [8].

## 5. Cyclooxygenase Pathway and Prostanoids in Glioblastoma Multiforme

### 5.1. Cyclooxygenase Pathway

Free PUFA, including ARA C20:4n-6, can be converted into prostanoids. This synthesis proceeds in two steps: the first reaction is catalyzed by COX: cyclooxygenase-1 (COX-1) and COX-2, whereas the second reaction is catalyzed by a prostanoid-specific synthase. The substrates for the production of prostanoids are dihomo-γ-linolenic acid C20:3n-6, ARA C20:4n-6, and EPA C20:5n-3, which are converted into 1-series [73], 2-series [73,74], and 3-series [75] prostaglandins or thromboxanes, respectively.

The most important prostanoids for tumorigenic processes in GBM are the 2-series prostanoids produced from ARA C20:4n-6. ARA C20:4n-6 is converted to prostaglandin G_2_ (PGG_2_) and then to prostaglandin H_2_ (PGH_2_) by COX [76,77,78], although during this reaction, the peroxygenated ARA C20:4n-6 can decompose with the generation of free radicals [79]. Cyclooxygenases also produce 9-hydroxyoctadecadienoic acid (9-HODE) from linoleic acid 18:2n-6 [80]. This compound is a ligand for PPARγ [81], transient receptor potential vanilloid 1 (TRPV1) [82], and G2A/GPR132 [83]; the latter is also a receptor for many lipid mediators produced in the LOX pathway.

COX-1 (another name is prostaglandin-endoperoxide synthase 1 (PTGS1)) is a constitutive enzyme with a constant level of expression [84]. A second enzyme with the same activity is COX-2 (another name is prostaglandin-endoperoxide synthase 2 (PTGS2)) [85], an inducible enzyme that is regulated at the transcriptional level and is characterized by rapid degradation of the COX-2 protein [86]. The half-life of the COX-2 protein is only 5 h.

Sometimes, cyclooxygenase-3 (COX-3), a variant of COX-1 that retains intron 1 in its mRNA, is also mentioned in the context of conversion to prostanoids [87]. Although there is expression of the COX-3 protein, which is longer than COX-1, this enzyme has the same activity as the other cyclooxygenases. In mice and dogs, COX-3 is more sensitive to the inhibitors acetaminophen and phenacetin. Humans also have a variant of COX-1, but it is as sensitive to these inhibitors as standard COX-1 [88].

PGH_2_ is unstable and undergoes spontaneous nonenzymatic conversion, mainly with PGE_2_ and, in smaller amounts, with prostaglandin D_2_ (PGD_2_) [78]. In the synthesis of PGE_2_, we can distinguish three synthases: membrane-bound prostaglandin E synthase-1 (mPGES-1)/*PTGES* [89,90,91], membrane-bound prostaglandin E synthase-2 (mPGES-2)/*PTGES2* [92], and cytosolic prostaglandin E synthase (cPGES)/*PTGES3* [93]. These synthases are dependent on glutathione, which serves to reduce the endoperoxide bridge in PGH_2_ with the formation of a single hydroxyl group. In addition, cPGES forms a complex with heat shock protein 90 (Hsp90), which is important in the activity of this PGE_2_ synthase [94]. mPGES-1 and mPGES-2 bind with either COX-1 or COX-2 [92,95,96], whereas cPGES binds only with COX-1 [93,97]. mPGES-1 is an inducible enzyme whose expression under the influence of inflammatory reactions increases following the expression of COX-2 [96]. mPGES-2 [96] and cPGES [93] are constitutive enzymes, meaning that their expression is not altered by inflammatory reactions.

In plasma, PGE_2_ undergoes enzymatic dehydration to PGA_2_ [98], which can isomerize to PGC_2_ via enzymes with PGA isomerase activity, and can then be isomerized to PGB_2_ via enzymes with PGC isomerase activity [98,99]. Importantly, detailed studies of the enzymes involved in these reactions are lacking.

PGH_2_ can also be enzymatically converted to other prostanoids by the appropriate synthase [97]. PGD_2_ is formed from this prostaglandin with the participation of hematopoietic-type prostaglandin D_2_ synthase (H-PGDS)/*HPGDS* and lipocalin-type prostaglandin D_2_ synthase (L-PGDS)/*PTGDS* [78]. It is also possible that pro-inflammatory prostaglandins are spontaneously converted into other prostaglandins with anti-inflammatory properties as a mechanism for regulating inflammatory responses [100].

PGD_2_ undergoes transformations to form the following prostaglandins: 15-deoxy-Δ^12,14^-PGD_2_ (15d-PGD_2_), PGJ_2_, Δ^12^-PGJ_2_, and 15d-PGJ_2_ [101,102]. PGD_2_ undergoes spontaneous non-enzymatic conversion to PGJ_2_ via dehydration or with Δ^15^-PGD_2_ [101,102]. PGJ_2_ can be spontaneously transformed directly into 15d-PGJ_2_ [102]. PGJ_2_ can be transformed with the participation of albumin into Δ^12^-PGJ_2_ [101,102,103,104].

As PGA_2_, PGJ_2_, 15d-PGJ_2_, and Δ^12^-PGJ_2_ have the same ring structure as cyclopentenone, they are classified as cyclopentenone prostaglandins [105]. Cyclopentenone prostaglandins have reactive electrophilic carbon atoms, which are responsible for the properties of this group of prostaglandins. These prostaglandins are inhibitors of nuclear factor κB (NF-κB) [6] and activators of PPARα and PPARγ [7,43]; thus, they have anti-inflammatory and anti-tumor properties.

It is possible that PGH_2_ can be converted to other prostanoids, such as TxA_2_ produced by thromboxane A synthase 1 (TBXAS1) [106,107]. TxA_2_ is unstable, as it undergoes non-enzymatic conversion to TxB_2_ with a TxA_2_ half-life of less than 40 s [108]; for this reason, TxA_2_ acts only locally at the site of synthesis.

TBXAS1 is responsible for the production of TxA_2_ and can also catalyze the conversion reaction of PGH_2_ into 12*S*-hydroxyheptadeca-5*Z*,8*E*,10*E*-trienoic acid (12-HHT) and malondialdehyde [106,109]. 12-HHT, produced by TBXAS1 in similar amounts to TxA_2_, is a ligand for leukotriene B_4_ receptor 2 (LTB_4_R2) [110,111,112].

PGH_2_ can be converted into PGI_2_ with PGIS (Figure 3) [113] or into PGF_2α_ with aldoketoreductase (AKR)1B1 and AKR1C3 [114,115]. PGF_2α_ can also be synthesized from PGE_2_ by AKR1C1 and AKR1C2 [114]. After synthesis, prostanoids are secreted outside the cell by multidrug resistance-associated protein 4 (MRP4)/ATP binding cassette subfamily C member 4 (ABCC4) [116].

Prostaglandins are first taken into the cell via prostaglandin transporter (PGT)/solute carrier organic anion transporter family member 2A1 (SLCO2A1), and they are inactivated and degraded [117,118]. Organic anions transporting polypeptide 3 (OATP3) and OATP4 are also involved in PGE_2_ uptake [119]. Then, prostaglandins are reduced by 15-hydroxyprostaglandin dehydrogenase (15-PGDH)/*HPGD* [118]. This reaction produces 15-keto-PGE_2_ from PGE_2_, a PPARγ ligand [120]. In a subsequent catabolic reaction, 15-keto-PGE_2_ is reduced by 12-hydroxyeicosanoid dehydrogenase (12-HEDH)/prostaglandin reductase 1 (PTGR1) [121] and prostaglandin reductase 2 (PTGR2) [120] through 15-oxoprostaglandin-Δ^13^-reductase (13-PGR) activity. This produces 13,14-dihydro-15-keto-PGE_2_ from 15-keto-PGE_2_.

Importantly, 13,14-dihydro-15-keto-PGE_2_ is unstable. It converts to 13,14-dihydro-15-keto-PGA_2_, and in this form, it combines with proteins, such as with albumin in plasma [122]. 13,14-dihydro-15-keto-PGA_2_ can also be converted to 11-deoxy-13,14-dihydro-15-keto-11,16-cyclo-PGE_2_ and occur in the blood in this form [122,123].

PGE_2_ can also be inactivated and degraded by β-oxidation [124,125]. It is first converted to PGE_2_-CoA [125], and then it is oxidized in peroxisomes and mitochondria, accompanied by the production of either dinor-PGE_2_ or tetranor-PGE_1_ [124].

PGE_2_ also undergoes ω-oxidation [126]. As a consequence of β-oxidation and ω-oxidation and also the action of 15-PGDH and PTGR1/2, 7α-hydroxy-5,11-diketotetranor-prosta-1,16-dioic acid is formed from PGE_2_, and then is excreted in the urine [127,128].

Acetylated COX-2 exhibits different catalytic properties than native COX-2. Although non-steroidal anti-inflammatory drugs (NSAID) prevent COX-2 catalytic activity, some NSAIDs cause acetylation of the COX-2 catalytic center. An example of such an NSAID is aspirin (acetylsalicylic acid), which causes changes in the catalytic properties of the enzyme. Acetylated COX-2 converts ARA C20:4n-6 into 15*R*-hydroxyeicosatetraenoic acid (15*R*-HETE) [129,130,131,132], whereas acetylated COX-1 has no catalytic activity [133].

Acetylated COX-2 also converts 5*S*-hydroxyeicosatetraenoic acid (5*S*-HETE) (the product of 5-lipoxygenase (5-LOX) activity) into 5*S*,15*R*-dihydroxyeicosatetraenoic acid (5S,15R-diHETE) [130,131]. Native COX-2 converts 5*S*-HETE into 5*S*,11*R*-diHETE, 5*S*,15*R*-diHETE, and 5*S*,15*S*-diHETE [130,131]. Then, 5-LOX converts 15*R*-HETE into 15-epi- lipoxin A_4_ (15-epi-LXA_4_) which has anti-inflammatory properties [134]. Another name for 15-epi-LXA_4_ is aspirin-triggered lipoxin (ATL). Acetylated COX-2 can also convert DHA C22:6n-3 and EPA C20:5n-3 into anti-inflammatory lipid mediators [5]. This means that aspirin has anti-inflammatory effects not only by inhibiting COX activity but also by causing the synthesis of lipid mediators with anti-inflammatory properties.

In addition to ARA C20:4n-6, dihomo-γ-linolenic acid C20:3n-6 and EPA C20:5n-3 are also converted with cyclooxygenases into 1-series prostaglandins [73] and 3-series prostaglandins [75], respectively. EPA C20:5n-3 reduces PGE_2_ production by COX-1 and, to a lesser extent, by COX-2 [135]. PGE_3_ binds to the same PGE_2_ receptors with less intracellular signal transduction efficiency [75]. PGE_3_ displaces PGE_2_ from the shared receptor, resulting in a decrease in the receptor’s activity. This means that PGE_3_ has anti-cancer properties.

PGE_1_ can also inhibit the proliferation of various cancer cells [136,137], although peroxidation of dihomo-γ-linolenic acid C20:3n-6 with COX-2 can result in the formation of PGH_1_ and the breakdown of the processed intermediate into free radicals [79]. COX-2 causes C-15 oxygenation of ARA C20:4n-6 and dihomo-γ-linolenic acid C20:3n-6. COX-2 can also catalyze C-8 oxygenation of dihomo-γ-linolenic acid C20:3n-6 [79,138], which often leads to the breakdown of the intermediate product and the formation of 8-hydroxyoctanoic acid (8-OH); this compound inhibits proliferation and is responsible for the antiproliferative properties of dihomo-γ-linolenic acid C20:3n-6 in cells with COX-2 expression [79,138], which is important for the inhibition of FADS1/D5D activity [139]. In the PUFA synthesis pathway, γ-linolenic acid C18:3n-6 in the acyl-CoA form is first elongated with Elovl5 to dihomo-γ-linolenic acid C20:3n-6 [14] and is then desaturated to ARA C20:4n-6 with FADS1/D5D. The reduction of FADS1/D5D activity results in the accumulation of dihomo-γ-linolenic acid C20:3n-6 in the cell. If such a cell has a high COX-2 expression, this fatty acid will either be converted into PGE_1_, or it will be broken during the reaction catalyzed by COX-2. This results in the formation of 8-OH-octanoic acid which inhibits tumor cell proliferation with a developed drug targeting FADS1/D5D activity [139].

### 5.2. Cyclooxygenase Pathway and Glioblastoma Multiforme

After ARA C20:4n-6 is released from cell membrane phospholipids, it is processed with COX and LOX. In the healthy brain, ARA C20:4n-6 is processed mainly with LOX, whereas in GBM tumors, it is processed mainly with COX, as shown by experiments on C6 cells [140].

COX-1 expression [141] and COX-2 expression [141,142] are elevated in GBM tumors compared to healthy brain tissue, whereas according to GEPIA and Seifert et al., just COX-1 expression is elevated [8,9]. The expression of all three PGE_2_ synthases, i.e., mPGES-1, mPGES-2, and cPGES, is also elevated in GBM [143], although according to GEPIA, only cPGES expression is higher compared to its expression in the healthy brain [9]. In contrast, Seifert et al. showed no change in PGE_2_ synthase expression in GBM tumors [8]. cPGES is enzymatically bound with just COX-1 [93,97]. Therefore, it is possible that COX-1-cPGES may play an important role in the production of PGE_2_ in GBM tumors. According to the GEPIA portal, there are also changes in the expressions of other prostaglandin synthases. In a GBM tumor, there is increased expression of H-PGDS but decreased expression of L-PGDS [9], both synthases involved in PGD_2_ synthesis. In contrast, Seifert et al. showed that the expression of H-PGDS and L-PGDS in a GBM tumor is lower than their expressions in a healthy brain [8]. According to GEPIA in a GBM tumor, there is also increased expression of AKR1B1, decreased expression of AKR1C1 and AKR1C2, and no change in AKR1C3 expression [9]. Similarly, Seifert et al. showed that in a GBM tumor, there is higher expression of AKR1B1 and decreased expression of AKR1C1, but there is no difference in the expressions of AKR1C2 or AKR1C3 between the GBM tumor and healthy brain tissue [8]. AKR1B1 is involved in the synthesis of PGF_2α_ [115], whereas AKR1C1 and AKR1C2 are involved in the conversion of PGE_2_ into PGF_2α_ [114]. Expression of the TxA_2_ synthesizing synthase TBXAS1 [8,9,144] is also elevated in GBM tumors, which may explain the increased expression and production of TxA_2_ and the higher TxA_2_/PGI_2_ ratio in GBM tumors than in healthy brain tissue [145,146].

As for receptors for prostaglandins, according to the GEPIA portal, there is an elevated expression of PTGER_4_/EP_4_ and TBXA_2_R/TP in the tumor relative to healthy brain tissue [9], these two being receptors for PGE_2_ and TxA_2_, respectively. In contrast, Seifert et al. showed that the expression of prostanoid receptors in GBM tumors did not differ relative to the healthy brain [8].

According to the GEPIA portal, the expression of MRP4/ABCC4 [9], a transporter responsible for the secretion of prostaglandins from the cell, is also increased in GBM tumors. The transcriptomics analysis by Seifert et al. did not confirm this [8]. GEPIA and Seifert et al. show no change in the expressions of PGT/SLCO2A1, 15-PGDH, 12-HEDH/PTGR1, and PTGR2 [8,9]—the first is a transporter that takes prostaglandins into the cell, and the second, third, and fourth are prostaglandin-degrading enzymes.

COX-2 is important in GBM tumor function. Its expression in GBM tumors is upregulated by hypoxia [23] and EGFR activation [147,148] as well as the action of epidermal growth factor receptor variant III (EGFRvIII) [147] and hepatocyte growth factor (HGF) [149]. COX-2 expression and biosynthesis of the most important product of this enzyme, PGE_2_, is present in GBM cancer cells. However, PGE_2_ in GBM tumors may not come mainly from GBM cancer cells but rather from tumor-associated macrophages (TAM) [150].

Under the influence of increased COX expression, there is increased production of PGE_2_, which is involved in tumorigenesis. PGE_2_ increases the expression of many factors relevant to tumorigenesis in GBM tumors—in particular, S100 calcium-binding protein A9 (S100A9) [151], interleukin 6 (IL-6) [152], and CXC motif chemokine ligand 8 (CXCL8)/interleukin 8 (IL-8) [153]. PGE_2_ also elevates proliferation [154,155,156] and causes migration of GBM cancer cells [156]. The effects on proliferation and migration are dependent on the receptors EP_2_ and EP_4_ [155,156], and perhaps also EP_3_. Activation of EP_3_ results in the activation of transient receptor potential melastatin 7 (TRPM7), which increases the proliferation and migration of GBM cells [157].

COX-2 is also important for GBM cancer stem cells. COX-2 expression, and with it, the production of PGE_2_, is higher in GBM cancer stem cells than in differentiated GBM cells [158,159]. This lipid mediator activates the Wnt pathway in GBM cancer stem cells, leading to the self-renewal and proliferation of these cells.

PGE_2_ induces angiogenesis in GBM tumors. Therefore, COX-2 expression is positively correlated with microvessel density in GBM tumors [160]. Notably, PGE_2_ causes vasculogenic mimicry of GBM cells, which promotes angiogenesis [161]. In GBM cells, PGE_2_ also increases the expression of CXCL8/IL-8 [153], which has pro-angiogenic properties [162].

PGE_2_ causes cancer immune evasion. Through EP_4_, PGE_2_ increases the expression of tryptophan-2,3-dioxygenase (TDO) [163], an enzyme that converts tryptophan into a signaling molecule that reduces immune cell activity.

PGE_2_ also affects tumor-associated cells which are important in cancer immune evasion. PGE_2_ increases the recruitment of myeloid-derived suppressor cells (MDSC) to the tumor niche in GBM [164] and interferes with the cytotoxic function of various immune cells, as shown by experiments in other cancer models. When acting chronically, PGE_2_ impairs the cytotoxic function of natural killer (NK) cells [165,166], dendritic cells [167], and T cells [168]. PGE_2_ also causes M2 polarization of macrophages [169], immunosuppressive cells that promote tumor growth.

PGE_2_ also causes radiation resistance [170,171] and TMZ resistance in GBM [172]. COX-2 expression and PGE_2_ production in GBM cancer cells are upregulated by TMZ [173] and ionizing radiation [170], which is related to caspase 3 activation in damaged cells and subsequent NF-κB activation [174]. Then, NF-κB increases COX-2 expression and, thus, the production of PGE_2_ that trans-activates EGFR and activates the β-catenin pathway, which has a pro-survival effect and leads to resistance to further therapy [170]. Through EP_1_ and EP_3_, PGE_2_ increases the intensity of β-oxidation and tricarboxylic acid cycle activity in mitochondria [172], leading to TMZ resistance. In response to ionizing radiation, healthy brain tissue also induces increased production of PGE_2_ and pro-inflammatory cytokines [175], which increases GBM cell migration as well as causes tumor recurrence [176].

PGD_2_ is also produced in GBM tumors [177]. At physiological concentrations, this prostaglandin increases the proliferation and migration of GBM cells, but, at concentrations of several micromoles, it decreases the viability and inhibits the proliferation of the GBM cells studied [177,178,179]. This effect may be due to 15d-PGJ_2_, which has anti-cancer properties [180]. PGD_2_ is non-enzymatically converted to 15d-PGJ_2_ [100]. High concentrations of PGD_2_ result in an accumulation of 15d-PGJ_2_ to a level that causes a measurable reduction in the viability of GBM cancer cells. Cyclopentenone prostaglandins, particularly PGJ_2_, Δ^12^-PGJ_2_, and 15d-PGJ_2_, have anti-tumor properties, as demonstrated in in vitro studies on GBM cells. These prostaglandins inhibit tumor cell proliferation through PPARγ activation [181,182].

TxA_2_ may also play an important role in tumorigenic processes in GBM. In GBM cells, TxA_2_ increases the expression of IL-6, which participates in tumorigenesis [183]. TBXAS1 inhibitors induce apoptosis and inhibit the migration and proliferation of GBM cancer cells [144,184,185], indicating an autocrine effect of TxA_2_. In addition, in an in vivo model, TBXAS1 inhibitors inhibited angiogenesis and GBM tumor growth [185]. The described inhibitors increased the sensitivity of GBM cells to alkylation chemotherapy [185] and radiotherapy [186].

Given the role of COX-2 in tumorigenesis in GBM, high COX-2 expression in GBM tumors is associated with poorer patient prognoses [160,187,188], although the GEPIA data showed no correlation between COX-1 and COX-2 expression and patient prognosis severity [9]. In addition, the expression of other prostanoid metabolism enzymes worsens the prognosis for GBM patients, in particular, high expression of mPGES-1, the synthase responsible for the production of PGE_2_ [121]. This is confirmed with the GEPIA data [9], although the expression levels of other PGE_2_ synthases are not associated with prognosis severity [9,121]. Of the other prostaglandin synthases, high expression of AKR1B1, a PGF_2α_-producing synthase [115], in GBM tumors is associated with poorer patient prognoses [9].

According to the GEPIA portal, expression of MRP4/ABCC4, a transporter that secretes prostaglandins from the cell, does not affect the prognosis for GBM patients [9]. Higher expression of certain prostaglandin receptors worsens the prognosis for patients with GBM. In particular, a worse prognosis is associated with higher expression of PTGER_1_/EP_1_ and PTGIR/IP [9], which are receptors for PGE_2_ and PGI_2_, respectively.

Higher expression of G2A/GPR132 is also associated with a worse prognosis (*p* = 0.052) in GBM patients [9]. G2A/GPR132 is a receptor for 9-HODE [83], a product of the activity of COX that process linoleic acid 18:2n-6 [80]. The role of this receptor in GBM has not been thoroughly investigated, although studies on fibroblasts have shown that G2A/GPR132 is an oncogene [189].

Prognosis severity is also affected by the expression level of enzymes involved in prostaglandin inactivation. High expression of 15-PGDH in GBM tumors is associated with a better prognosis [121]. The opposite is true for the expression of 12-HEDH/PTGR1, the enzyme that catalyzes the second prostaglandin inactivation reaction [121]. PGT/SLCO2A1 expression levels are not associated with prognosis severity. On the other hand, according to the GEPIA portal, the expressions of PGT/SLCO2A1, 15-PGDH, PTGR1, and PTGR2 do not affect the prognosis for patients with GBM [9].

Prostaglandin levels in GBM tumors may also be associated with a worse prognosis, particularly higher levels of PGE_2_ and PGF_2α_ [121]. PGD_2_ levels in GBM tumors do not affect prognosis severity [121]. At the same time, these lipid mediators are often unstable, transforming into other lipid mediators with lesser or different properties within a short time after synthesis. For this reason, they may act locally in the immediate vicinity of the site of their synthesis.

Relating enzyme expression and levels of the discussed prostaglandins to prognosis makes it possible to estimate the significant impact of a particular pathway on cancer processes. In GBM tumors, higher expressions of production enzymes and levels of PGE_2_ (COX-2, mPGES-1) and PGF_2α_ (COX-2, AKR1B1) are responsible for worse prognoses [9,121]. On the other hand, higher expression of the prostaglandin-inactivating enzyme, 15-PGDH, is associated with better prognoses (Table 5) [121]. For this reason, NSAIDs are being investigated as either potential drugs [190,191] or agents with chemopreventive properties against GBM. Various meta-analyses inconclusively discuss the chemopreventive properties of NSAIDs, such as aspirin. Depending on the meta-analyses cited, regular use of NSAIDs, including aspirin, may either reduce the risk [192,193] or have no effect [194] on the risk of developing glioma or GBM. Nevertheless, the COX pathway produces prostaglandins that exhibit pro-cancer and anti-cancer properties. A better option may be to develop drugs that specifically target only particular prostaglandins relevant to tumorigenic processes in GBM, namely PGE_2_ and PGF_2α_ [9,121]. It may be possible to develop drugs that are specific inhibitors of mPGES-1.

### 5.3. Pan-Cancer Analysis of Genes Related to the COX Pathway and GBM

Changes in the expression of various genes in GBM tumors relative to healthy tissue may be the result of tumor-specific neoplastic processes or specific mechanisms found only in GBM. For this reason, we performed a pan-cancer analysis of the expression of the genes involved in the COX pathway based on the data available in the GEPIA web server [9]. It showed that increased or decreased expression of a given gene relative to healthy tissue does not occur in all types of cancer. At the same time, in some cases, a certain trend of changes in the expressions of the genes studied can be observed. An example of this is *TBXAS1,* whose expression is increased in nine types of cancer but decreased in another four types of cancer. Similarly, the expression of mPGES-1/PTGES is increased in eight types of cancers but decreased in three types of cancers. Some genes tend to undergo decreased expression in tumors. An example of this is 15-PGDH*/HPGD,* whose expression is reduced in 18 types of cancer but increased in two types of cancer relative to healthy tissue. Another example is the expression of PGIS/*PTGIS*, decreased in 17 types of cancer but elevated only in pancreatic adenocarcinoma.

According to GEPIA, there is an increase in COX-1*/PTGS1* expression in GBM tumors, which is the same as in lower grade glioma. In seven types of tumors, this gene is overexpressed, but in seven more types, its expression is reduced. This indicates that the increased expression of COX-1/*PTGS1* in gliomas (GBM and lower grade gliomas) is specific to these diseases. Some studies also show increased expression of PGE_2_ synthases (mPGES-1/*PTGES*, mPGES-2/*PTGES2* and cPGES/*PTGES3*) [9,143], although GEPIA confirms it is only for cPGES/*PTGES3* [9]. According to GEPIA, in lower grade glioma, there are no changes in the expression of PGE_2_ synthases relative to healthy brain tissue. According to GEPIA, expression of cPGES/*PTGES3* is increased in 11 types of tumors but is decreased in one type. For this reason, the increase in cPGES/*PTGES3* expression in GBM can be considered cancer-specific, just like mPGES-1/*PTGES*, which has increased expression in eight types of cancer and decreased in three. According to GEPIA, only four types of cancers have increased expression of mPGES-2/*PTGES2*, which shows that this enzyme may not be cancer-specific.

According to GEPIA in GBM, there is also increased expression of H-PGDS/*HPGDS* but decreased expression of L-PGDS/*PTGDS* [9]. At the same time, Seifert et al. showed that the expression of both PGD_2_ synthases is decreased in GBM tumors [8]. H-PGDS/*HPGDS* expression is also upregulated in lower grade glioma. H-PGDS/*HPGDS* expression is downregulated in five tumor types and upregulated in an equal number of tumor types. Changes in H-PGDS/*HPGDS* expression can be specific to gliomas. L-PGDS/*PTGDS* expression is lower in GBM compared to healthy brain tissue [8,9]. L-PGDS/*PTGDS* expression is decreased in almost all types of tumors and, thus, can be deemed specific to cancer.

In GBM, as in lower grade glioma, there is increased expression of *TBXAS1* [8,9,144]. The expression of this enzyme is elevated in nine types of tumors, which means it may be cancer-specific.

In GBM tumors, there is also upregulation of *AKR1B1* expression but downregulation of *AKR1C1* and *AKR1C2* expressions relative to healthy brain tissue [8,9]. Lower grade gliomas show no changes in the expressions of these enzymes. The expression of *AKR1B1* increases in nine types of tumors. *AKR1C1* and *AKR1C2*, on the other hand, have decreased expressions in 14 types of tumors, which indicates that these changes may be cancer-specific.

PGIS/*PTGIS* expression is downregulated in 17 types of tumors. At the same time, in GBM tumors, PGIS/*PTGIS* expression does not differ from healthy brain tissue [8,9].

In GBM and lower grade glioma, there is an increase in MRP4/*ABCC4* expression [9]. This transporter also has increased expression in another four types of tumors but decreased expression in two types of tumors. Changes in MRP4/*ABCC4* expression may be specific to gliomas.

Finally, 15-PGDH/*HPGD* expression is often downregulated in tumors (Table 6). This was shown by a pan-cancer analysis in which 18 out of 31 cancers had decreased expression of this enzyme. At the same time, in gliomas (GBM and lower grade glioma), there were no changes in 15-PGDH/*HPGD* expression relative to healthy brain tissue.

## 6. Lipoxygenases and Arachidonic Acid in Glioblastoma Multiforme

### 6.1. Lipoxygenases Pathway

In addition to the COX pathway, PUFA can be transformed with LOX. These enzymes exhibit dioxygenase activity, catalyzing the insertion of a hydroperoxyl group into a PUFA, most commonly ARA 20:4n-6. Hydroperoxyeicosatetraenoic acids (HpETE) are then formed from ARA 20:4n-6, which are further processed in the lipoxygenase pathway. The names of LOX enzymes are related to their sites of formation and the configuration of the hydroperoxyl group in ARA 20:4n-6. In humans, there are six LOX:epidermal lipoxygenase 3/arachidonate lipoxygenase 3 (eLOX3/*ALOXE3*),5-lipoxygenase/arachidonate 5-lipoxygenase (5-LOX/*ALOX5*),12S-lipoxygenase/arachidonate 12-lipoxygenase, 12S type (12S-LOX/*ALOX12*),12R-lipoxygenase/arachidonate 12-lipoxygenase, 12R type (12R-LOX/*ALOX12B*),15-lipoxygenase-1/arachidonate 15-lipoxygenase (15-LOX-1/*ALOX15*), also known as 12/15-LOX, and15-lipoxygenase-2/arachidonate 15-lipoxygenase type B (15-LOX-2/*ALOX15B*).

The *ALOX5* gene is found on chromosome 10. The other LOX form a gene cluster on 17p13.1 [195,196]. There is also a mouse 8-LOX [197], whose sequence is 78% identical to that of human 15-LOX-2/*ALOX15B* [197,198]. It is likely that mouse 8-LOX and human 15-LOX-2/ALOX15B are derived from a common ancestor, which was indirectly confirmed by mutagenesis experiments on these two enzymes. Changing only two amino acids in either mouse 8-LOX or human 15-LOX-2/*ALOX15B* alters the catalytic properties of these two enzymes in 15-LOX and 8-LOX, respectively [197].

#### 6.1.1. Epidermal Lipoxygenase 3

The *ALOXE3* gene forms a gene cluster on 17p13.1 together with other LOX [196]. The highest expression of the *ALOXE3* gene is found in the skin [196,199]; very low expression of this gene is found in the brain, placenta, pancreas, ovary, and testis.

eLOX3/*ALOXE3* shows no significant activity against ARA 20:4n-6 or linoleic acid C18:2n-6 [200], which is related to the low availability of molecular oxygen in the active center of this enzyme [201]. For this reason, the processing of ARA 20:4n-6 by eLOX3/*ALOXE3* is very inefficient, but eLOX3/*ALOXE3* can exhibit dioxygenase activity to ARA 20:4n-6.

eLOX3/*ALOXE3* has hydroperoxide isomerase activity [200]. eLOX3/*ALOXE3* converts HpETE into hydroxy-epoxyeicosatrienoic acid, which is the main product of eLOX3/*ALOXE3* activity. eLOX3/*ALOXE3* also converts HpETE into oxo-eicosatetraenoic acid (oxo-ETE)/ketoeicosatetraenoic acid (KETE) [200,202]. 15S-HpETE is converted by eLOX3/ALOXE3 into either 13*R*-hydroxy-14*S*,15*S*-epoxyeicosa-5*Z*,8*Z*,11*Z*-trienoic acid or 15-oxo-ETE [200].

eLOX3/*ALOXE3* also converts 12*S*-HpETE into hepoxilin A_3_ (HxA_3_), HxB_3_ [200,203], or 12-oxo-ETE [204,205]. On the other hand, 12*R*-HpETE is converted by eLOX3/*ALOXE3* into either 11,12-bis-epi-HxA_3_ or 12-oxo-ETE [200].

In addition, eLOX3/*ALOXE3* shows activity to 5-HpETE and other HpETEs [202]. Because HETE and oxo-ETE [206] as well as hepoxilins [207] exhibit biological activity, eLOX3/*ALOXE3* affects biological and pathological processes, particularly in the skin, where expression of this enzyme is highest. For this reason, mutations in the *ALOXE3* gene lead to ichthyosis [208,209,210].

#### 6.1.2. 5-Lipoxygenase

The best-studied LOX is 5-LOX/*ALOX5*. The highest expression of 5-LOX/*ALOX5* is found in the bone marrow, appendix, lung, urinary bladder, spleen, and lymph node [199]. This enzyme converts ARA 20:4n-6 to 5*S*-hydroperoxyeicosatetraenoic acid (5-HpETE) and then to leukotriene A_4_ (LTA_4_) [211]. Importantly, 5-lipoxygenase-activating protein (FLAP)/*ALOX5AP* is required for the activity of 5-LOX/*ALOX5*. FLAP/*ALOX5AP* is a substrate carrier [212,213]. 5-HpETE is an activator of PPARα [214]; for this reason, if it is not converted to other lipid mediators, then it will activate this nuclear receptor. Subsequently, LTA_4_ is converted to other lipid mediators, in particular to other leukotrienes. LTA_4_ can also undergo spontaneous conversion to 5,6-diHETE, 5,12-diHETE, and 5-oxo-ETE [215]. In turn, 5-HpETE is converted to 5-hydroxyeicosatetraenoic acid (5-HETE) with glutathione peroxidase [216]. The identified receptor for 5-HETE is G2A/GPR132 [83]; this receptor is also activated by other lipid mediators, such as various HETE and 9-HODE.

5-oxo-ETE can also be formed from 5-HETE with the participation of an enzyme with 5-hydroxyeicosanoid dehydrogenase (5-HEDH) activity [217,218,219]. 5-oxo-ETE is an important lipid mediator with a receptor oxoeicosanoid receptor 1 (OXER1)/GPR99 [220,221,222].

LTA_4_ is a precursor for the production of other leukotrienes and lipoxins; it is converted to lipoxins in a reaction catalyzed by 12-LOX or 15-LOX [223]. LTA_4_ can also be converted to LTB_4_ by LTA_4_ hydrolase (LTA_4_H) [224,225]. LTA_4_H also has aminopeptidase activity unrelated to the production of leukotrienes [225]; this activity is important in moderating the immune response [226]. LTB_4_ has its own membrane receptors: LTB_4_R1/BLT1 [227] and LTB_4_R2/BLT2 [228]. Inside the cell, LTA_4_ and LTB_4_ activate PPARα, by which these leukotrienes can exert anti-inflammatory effects [7,214].

Glutathione can be attached to LTA_4_ by LTC_4_ synthase (LTC_4_S) (Figure 4) [229,230]. LTC_4_ is then formed. LTC_4_S combines with 5-LOX and FLAP to increase the efficiency of LTC_4_ production with ARA 20:4n-6 [231]. Subsequently, amino acids from the conjugated glutathione in LTC_4_ can be removed. As a consequence of this, LTC_4_ is converted into other leukotrienes, namely LTD_4_, LTE_4_, and LTF_4_. All of these leukotrienes, together with LTC_4_, form a group called cysteinyl leukotrienes. LTD_4_ is then formed from LTC_4_ with the involvement of γ-glutamyltransferase 1 (GGT1) and γ-glutamyltransferase 5 (GGT5) [232]. Subsequently, LTD_4_ can be converted to LTE_4_ with the participation of dipeptidase 1 (DPEP1) and dipeptidase 2 (DPEP2) [233,234]. LTC_4_ can also be converted to LTF_4_ with the participation of carboxypeptidase A [235]. Amino acids can be attached back to cysteine in cysteinyl leukotriene, as exemplified by the conversion of LTE_4_ to LTF_4_ with the participation of an enzyme with γ-glutamyltranspeptidase activity [236]. LTF_4_, however, has a much weaker effect than LTE_4_, and the latter reaction can be considered an inactivation of LTE_4_.

Once synthesized, leukotrienes are secreted from the cell. LTC_4_ is secreted from cells by multidrug resistance-associated proteins (MRP) [237]. In particular, MRP1/ABCC1 [238,239], MRP2/ABCC2 [240,241], MRP3/ABCC3 [242], MRP4/ABCC4 [243], MRP6/ABCC6 [244], MRP7/ABCC10 [245], and MRP8/ABCC11 [246] are responsible for this process. In contrast, OATP1/SLCO1C1 and OATP4 are responsible for the uptake of LTC_4_, particularly into liver cells where leukotrienes are degraded [119,247]. In contrast, LTB_4_ transport is still poorly studied; it is known that efflux of LTB_4_ occurs via MRP4/ABCC4 [243].

Once leukotrienes are secreted outside the cells, they can activate their membrane receptors. LTB_4_ has two receptors: LTB_4_R1/BLT1 [227] and LTB_4_R2/BLT2 [228], the former of which has a 20 times better dissociation constant (Kd) than LTB_4_R2 in binding LTB_4_ [228]. With that said, LTB_4_R2 can be activated by other ARA-derived lipid mediators. These include 12*S*-HETE, 12*R*-HETE, 15-HETE, 15-HpETE [248], and 12-HHT [110,111,112]. 12-HHT is formed together with malondialdehyde in a reaction catalyzed by TBXAS1, whose substrate is PGH_2_ [106,109]. In addition, 12-HHT can be formed independently of TBXAS1 but in smaller amounts [109].

The receptors for cysteinyl-leukotrienes are CysLTR_1_ [249] and CysLTR_2_ [250,251]. Both receptors show a 38% similarity in amino acid sequence [250]. CysLTR_1_ shows a high affinity for LTD_4_ and low affinity for LTC_4_ and LTE_4_, and it shows no affinity at all for LTB_4_ [249]. CysLTR_2_ has the best affinity for LTC_4_ and LTD_4_ and a very low affinity for LTE_4_, and it shows no affinity at all for LTB_4_ [250,251]. A receptor specific for LTE_4_ is 2-oxoglutarate receptor 1 (OXGR1)/GPR99 [252], which is also the receptor for 2-oxoglutarate. This receptor has a lower affinity for LTC_4_ and LTD_4_. Another identified receptor for cysteinyl-leukotrienes specifically for LTC_4_ and LTD_4_ is G protein-coupled receptor 17 (GPR17) [253], which is also activated by uridine diphosphate (UDP), UDP-glucose, and UDP-galactose [253]. Further studies have not confirmed that GPR17 is a receptor for UDP, LTC_4_, and LTD_4_ [254,255]. This receptor can, independently of its ligand, downregulate CysLTR_1_ [256], which means it can reduce the action of cysteinyl leukotrienes.

Leukotrienes can be inactivated and excreted. LTB_4_ is oxidized to 12-oxo-LTB_4_ with 12-hydroxyeicosanoid dehydrogenase (12-HEDH)/PTGR1 [257,258,259]. This enzyme is also involved in prostaglandin degradation [121]. Subsequently, 12-oxo-LTB_4_ is reduced with the formation of 12-oxo-10,11-dihydro-LTB_4_ with an enzyme with Δ^10^-reductase activity [260]. 12-oxo-10,11-dihydro-LTB_4_ can then be converted to 10,11-dihydro-LTB_4_ and 10,11-dihydro-12-epi-LTB_4_, which undergo ω-oxidation, β-oxidation, or elongation [257]; compounds formed after ω-oxidation and β-oxidation are excreted in the feces [261] and urine [262] as ω-carboxymetabolites of LTB_4_. HETE are similarly degraded, such as 12-HETE with the formation of 10,11-dihydro-12-HETE and 10,11-dihydro-12-oxo-ETE [263]. Cysteinyl-leukotrienes are first converted to LTE_4_ [264]; this leukotriene then undergoes ω-oxidation with the formation of ω-carboxy-tetranor-dihydro-LTE4, which is eliminated in the feces and urine.

#### 6.1.3. 12S-Lipoxygenase

*ALOX12* gene expression is found in the esophagus and skin [199]. 12S-LOX/*ALOX12* can participate in the conversion of LTA_4_ into lipoxins [223], but the best-described activity of 12S-LOX/*ALOX12* is to catalyze the insertion of a hydroperoxyl group into ARA 20:4n-6 at position 12—12*S*-HpETE is then formed [265]—the compound which can also be formed with 15-LOX-1/*ALOX15* [266].

12S-LOX can convert dihomo-γ-linolenic acid to 12*S*-hydroxy-8*Z*,10*E*,14*Z*-eicosatrienoic acid (12*S*-HETrE) [267,268]. In contrast, linoleic acid C18:2n-6 is not a substrate for 12S-LOX/*ALOX12* [267]. 12*S*-HpETE can be converted to 12*S*-HETE, whose receptors are G protein-coupled receptor 31 (GPR31) [269] and G2A/GPR132 [83].

12*S*-HETE also activates PPARγ [270], as 12*S*-HpETE [200] and 12*S*-HETE can be converted to 12-oxo-ETE [260], a PPARγ ligand and activator [204]. 12-oxo-ETE can be converted back to 12*S*-HETE with an enzyme with 12-oxo-ETE reductase activity [271].

12*S*-HpETE can be converted to HxA_3_ (8-hydroxy-11,12-epoxyeicosatrienoic acid) or HxB_3_ (10-hydroxy-11,12-epoxyeicosatrienoic acid) with enzymes with hepoxilin synthase activity, for example, heme, as shown by experiments on hemoglobin and hemin [272,273]. Hepoxilin synthase activity is also demonstrated by eLOX3/*ALOXE3*, 12S-LOX/*ALOX12*, and 15-LOX-1/*ALOX15*, as shown by experiments on human, rat, and mouse models [200,203,274,275].

Then, HxA_3_ may bind glutathione via glutathione S-transferase at position 11 [276,277]. HxA_3_ then gives rise to 11-glutathionyl-HxA_3_, or otherwise HxA_3_-C. HxB_3_ is not subject to such modification [278]. HxA_3_-C can be produced in the brain and may be a neuromodulator [279]. Like cysteinyl-leukotrienes, HxA_3_-C can be converted to other cysteinyl-hepoxilins [279]. HxA_3_-C is converted to HxA_3_-D by γ-glutamyltranspeptidase. HxA_3_ and HxB_3_ can also be converted into trioxilin A_3_ (TrXA_3_) (8,11,12-trihydroxyepoxyeicosatrienoic acid) and TrXB_3_ (10,11,12-trihydroxyepoxyeicosatrienoic acid) with soluble epoxide hydrolase (sEH) (current name: epoxide hydrolase 2 (EPHX2)) [276,280]. HxA_3_ receptors are TRPV1 and transient receptor potential ankyrin 1 (TRPA1) [281,282]. HxA_3_ and TrXA_3_ are also antagonists of the TP receptor [283], the receptor for TxA_2_.

#### 6.1.4. 12R-Lipoxygenase

In addition to 12S-LOX/*ALOX12*, there is a second enzyme with 12-LOX activity [195], namely 12R-LOX/*ALOX12B* [284]. This enzyme shows activity towards ARA C20:4n-6 but not linoleic acid C18:2n-6 [284]. 12R-LOX/*ALOX12B* transforms ARA C20:4n-6 into 12*R*-HpETE, a stereoisomer of the product of 12S-LOX/*ALOX12*’s enzyme activity. 12*R*-HpETE is converted to 11,12-bis-epi-HxA_3_ with eLOX3/*ALOXE3* [200]. 12*R*-HpETE is a stereoisomer of 12*S*-HpETE. Similar to this compound, 12*R*-HpETE can also be converted to 12*R*-HETE [206], which is then converted to 12-oxo-ETE with an enzyme with 12-hydroxyeicosanoid dehydrogenase activity [206,260], including eLOX3/*ALOXE3* [200].

The *ALOX12B* gene is only 38% similar to the *ALOX12* gene. The highest expression of this enzyme is found in the skin, and it is much lower in the prostate and adrenal gland [196,199,284]. 12R-LOX is important in skin function; mutations in the *ALOX12B* gene lead to ichthyosis [208,210,285], as do mutations in the *ALOXE3* gene. 12R-LOX/*ALOX12B* and eLOX3/*ALOXE3* participate in a common pathway in lipid mediator production. 12R-LOX produces 12*R*-HpETE, which is converted to 11,12-bis-epi-HxA_3_ with eLOX3 (Figure 5) [200]. Under the influence of eLOX3/*ALOXE3*, 12-oxo-ETE is also formed from 12*R*-HpETE in small amounts [200].

#### 6.1.5. 15-Lipoxygenases

Like the previously described LOX, 15-LOX catalyzes the formation of 15*S*-hydroperoxyeicosatetraenoic acids (15-HpETE) from ARA 20:4n-6 [286]. In humans, two 15-LOX isoforms are distinguished: 15-LOX-1/*ALOX15* [287] and 15-LOX-2/*ALOX15B* [288]. The highest expression of 15-LOX-1/*ALOX15* is found in the lung, and the lower expressions are in the skin, intestine, heart, lymph node, and testis [199]. The highest expression of 15-LOX-2/*ALOX15B* is found in the prostate and skin. Expression of this enzyme is also observed in the lung, esophagus, and cornea [196,199,288].

The enzymatic properties of the two isoforms differ. 15-LOX-1/*ALOX15* catalyzes the formation of 15-HpETE, but it also converts part of the substrate, ARA 20:4n-6, into 12-HpETE [266]—for this reason, the enzyme owns its historical name: 12/15-LOX. 15-LOX-2/*ALOX15B* has no such activity [266,288].

15-LOX-1/*ALOX15* shows much higher activity with linoleic acid C18:2n-6 than 15-LOX-2/*ALOX15B* (Figure 6) [266]. These enzymes convert linoleic acid C18:2n-6 into 13*S*-hydroperoxyoctadecadienoic acid (13-HpODE), which converts to 13*S*-hydroxyoctadecadienoic acid (13-HODE). The identified receptor for 13-HpODE is G2A/GPR132 [83]. 13-HODE also activates the TRPV1 receptor [82]. 13-HODE undergoes the same transformations as HETE and can be oxidized to 13-oxo-ODE. 13-oxo-ODE [289] and 13-HODE [290] are PPARγ ligands.

15-HpETE is transformed into many lipid mediators. It can be transformed into 15-HETE, which is an activator of PPARγ [270] and G2A/GPR132 [83]. 15-HpETE can be converted to 13*R*-hydroxy-14*S*,15*S*-epoxyeicosa-5*Z*,8*Z*,11*Z*-trienoic acid (14,15-HxB_3_ 13*R*), 11*S*-hydroxy-14*S*,15*S*-epoxy-5*Z*,8*Z*,12*E*-eicosatrienoic acid (14,15-HxA_3_ 11*S*), and 15-oxo-ETE [200,291]. 14,15-HxA_3_ 11*S*, analogous to HxA_3_, can be conjugated with glutathione. This produces 14,15-HxA_3_-C 11*S* and cysteinyl-14,15-HxA_3_ 11*S,* having conjugated glutathione without further amino acids, which is analogous to that of cysteinyl-leukotriene [291].

15-HpETE can also be converted to eoxins [292], which are isomers of leukotrienes.

15-HpETE can also be converted to lipoxins with 5-LOX [223], resulting in the formation of 5*S*,15*S*-dihydroperoxyeicosatetraenoic acid (5,15-diHpETE), and then converted to LXA_4_ or LXB_4_ [293]. 5-HpETE can also be converted with 15-LOX-1/*ALOX15* into 5,15-diHpETE and, via the same pathway, be converted into LXA_4_ or LXB_4_ [293]. 15-HETE can be converted to LXA_4_ with 5-LOX/*ALOX5* [294]. Lipoxins can also be formed from LTA_4_, which is processed by 15-LOX-1/*ALOX15* or 12-LOX [293,295].

LXA_4_ is a lipid mediator with biological activity whose receptors are lipoxin A_4_ receptor (ALX)/formyl peptide receptor type 2 (FPR2) [296,297], aryl hydrocarbon receptor (AHR) [298], and estrogen receptors subtypes alpha (ERα) [299], the former of which is not a receptor for LXB_4_ [296]. The ALX/FPR2 receptor is responsible for the anti-inflammatory properties of lipoxins.

There are also cysteinyl lipoxins, which, just like cysteinyl leukotrienes, are lipoxins with conjugated glutathione at carbon 6 [294]. They are synthesized from 15-HETE, from which, with the participation of 5-LOX/*ALOX5*, 15-hydroxy-5,6-epoxy-eicosatetraenoic acid is formed, a compound similar in structure to LTA_4_. The epoxy group from these two compounds is converted to a hydroxyl group and conjugated glutathione [294]. However, it is not known whether cysteinyl lipoxins are essential lipid mediators or merely arise as a result of the nonspecificity of enzymes conjugating glutathione to various compounds.

### 6.2. Lipoxygenases in Glioblastoma Multiforme

In GBM tumors, ARA C20:4n-6 is mainly processed by COX, as shown by experiments on the C6 cell line [140]. In contrast, in the healthy brain, this PUFA is mainly processed by the LOX pathway. This shows that in GBM tumors, the LOX pathway may not be as important as the COX pathway, although it is still important in tumor mechanisms in GBM tumors.

#### 6.2.1. 5-Lipoxygenase Pathway in Glioblastoma Multiforme

The expression of 5-LOX/*ALOX5* in a GBM tumor is higher than in non-tumor brain tissue [300,301,302]. This is also confirmed by data obtained from the GEPIA portal [9] and from Seifert et al. transcriptomics analysis [8].

Expression of 5-LOX/*ALOX5* in the GBM tumor is found in macrophage and microglial cells as well as in other cells, such as cancer cells [301,302]. It is higher in GBM cancer stem cells than in other GBM cancer cells [303]. According to GEPIA, higher expressions of FLAP/*ALOX5AP*, LTC_4_S, LTA_4_H, GGT5, and DPEP1 but not DPEP2 [9], the enzymes that synthesize LTB_4_ and LTE_4_ from the product of 5-LOX/*ALOX5* activity, were also found in GBM tumors [224,225,229,230,232,234]. Seifert et al. showed that there are higher expressions of FLAP/*ALOX5AP*, LTA_4_H, and GGT5 in GBM tumors than in healthy brain tissue [8]. In contrast, LTC_4_S, DPEP1, and DPEP2 are not affected. The higher expression of enzymes responsible for leukotriene biosynthesis increases the production [304] and levels [305] of these lipid mediators further in GBM tumors than in healthy brain tissue, particularly cysteinyl-leukotrienes.

The expression level of 5-LOX/*ALOX5* in GBM tumors does not affect prognosis [9,188], although simultaneous high expression of COX-2 and 5-LOX/*ALOX5*, two major ARA C20:4n-6 processing enzymes, is associated with a worse prognosis [188]. This shows that the two pathways in cooperation can impinge on prognosis severity.

The expression levels of most enzymes involved in leukotriene production and metabolism do not affect prognosis [9]. Only for GGT1, higher expression in GBM tumors is associated with a worse prognosis [9]. GGT5 expression showed a positive trend (*p* = 0.055) toward a worse prognosis. GGT1 and GGT5 are enzymes that catalyze the transformation of LTC_4_ into LTD_4_ [232], demonstrating that the transformation of cysteinyl leukotrienes may be important in tumorigenesis in GBM.

In addition, higher expression of 12-HEDH/PTGR1, an enzyme that degrades LTB_4_, as well as prostaglandins, may be associated with worse prognoses for GBM patients [121], although GEPIA did not confirm such a link [9]. In addition, GEPIA and Seifert et al. did not show that 12-HEDH/PTGR1 expression differs between GBM tumors and healthy brain tissue [8,9]. According to GEPIA [9] and Seifert et al. [8], expression levels of receptors for leukotrienes LTB_4_R1, LTB_4_R2, CysLTR_1_, CysLTR_2_, GPR17, and OXGR1/GPR99 do not differ between GBM tumors and healthy brain tissue. In addition, the expression levels of these receptors in GBM tumors do not affect prognosis [9].

Leukotrienes as well as the entire 5-LOX pathway are important in tumorigenesis in GBM. They may also be important in the onset of GBM and in the first stages of tumorigenesis. The GA genotype of rs2291427 in the *ALOX5* gene is associated with a higher risk of GBM in men [306].

Expression of 5-LOX/*ALOX5* is higher in GBM cancer stem cells than in other GBM cancer cells [303]. The products of 5-LOX/*ALOX5* activity induce proliferation and self-renewal of GBM cancer stem cells. The effects of 5-LOX/*ALOX5* on GBM cancer stem cells are autocrine in nature.

LTB_4_ also increases the proliferation of GBM cells [307]. This is associated with an increase in Ca^2+^ levels in the cytoplasm of GBM cells [307]. Studies of various cell lines show that 5-LOX/*ALOX5* expression is present in only a portion of them [308,309]. Expression of 5-LOX/*ALOX5* causes an autocrine increase in the proliferation of such a line and, thus, makes culture growth dependent on 5-LOX/*ALOX5* activity. All GBM lines express LTA_4_H, LTB_4_R1/BLT1, LTB_4_R2/BLT2, and CysLTR_2_, but only some lines express LTC_4_S [309], indicating heterogeneity in the production of cysteinyl-leukotrienes and 5-HETE by GBM cancer cells.

The dependence of the proliferation of some GBM cancer cell lines on the 5-LOX pathway may be a potential therapeutic target for GBM treatment in personalized therapy. For this reason, the pan-LOX inhibitor Nordy [303,310], 5-LOX inhibitors such as caffeic acid [307], A861 [311], AA-863, and U-60,257 (pyriprost) [312], LTA_4_H inhibitors such as bestatin [311], and CysLTR_1_ and CysLTR_2_ receptor inhibitors such as montelukast and zafirlukast [313] have anti-tumor properties against GBM and inhibit proliferation. This is associated with decreased ERK MAPK activation and induction of apoptosis as a result of decreased expression of anti-apoptotic Bcl-2 and increased expression of pro-apoptotic Bax [308].

Cysteinyl leukotrienes may have anticancer properties by increasing the bioavailability of various chemotherapeutics. In the brain, as well as in GBM tumors, there is a blood-brain barrier (BBB) that is poorly permeable to many substances, including anticancer drugs [314]. However, cysteinyl leukotrienes have BBB permeability, as shown by experiments on rat RG-2 glioma tumors [315]. BBB permeability is highest for LTE_4_ [315], with cysteinyl leukotrienes not causing BBB permeability in healthy brain tissue [315,316]. For this reason, the administration of LTC_4_ prior to the administration of chemotherapeutics that pass poorly through the BBB increases the bioavailability of drugs such as cisplatin [317]. However, this method does not increase the bioavailability of all chemotherapeutics, as exemplified by paclitaxel [318].

The receptor for cysteinyl leukotrienes is GPR17 [253]. According to GEPIA [9] and Seifert et al. [8], the expression level of this receptor does not differ between GBM tumors and healthy brain tissue. Higher GPR17 expression is associated with better prognosis in patients with low-grade gliomas, according to the Chinese Glioma Genome Atlas (CGGA) [319] and GEPIA [9], but the expression of this receptor is not associated with prognosis in a GBM patient [9]. GPR17 expression is also higher in low-grade gliomas than in healthy brain tissue [319]. Activation of this receptor by the ligand inhibits proliferation in the G_1_ phase and induces apoptosis of GBM cell lines LN-229 and SNB-19 [319]. In addition, GPR17 ligands inhibit tumor growth, as shown by experiments using patient-derived xenograft mouse models. The action of GPR17 is associated with a decrease in the levels of cyclic adenosine monophosphate (cAMP) and Ca^2+^ in the cytoplasm, which reduces the activation of the PI3K → Akt/PKB pathway [319,320]. An increase in GPR17 expression can cause the proliferation and migration of GBM cells [321], particularly with an increase in the expression of this receptor by long non-coding RNA (lncRNA) colorectal neoplasia differentially expressed (CRNDE) in low-grade glioma cells [321].

The receptor for 5-HETE, and also other lipid mediators, is G2A/GPR132 [83]. Higher expression of this receptor, according to GEPIA, is associated with a worse prognosis for a GBM patient (*p* = 0.052) [9], yet there is no significant upregulation of this receptor expression in GBM tumors [8,9].

5-oxo-ETE may also play an important role in tumorigenic mechanisms in GBM. The receptor for this lipid mediator is OXER1/GPR99 [220,221,222]. The expression of this receptor does not differ between GBM tumor and healthy brain tissue [8,9]. According to GEPIA, higher expression of OXER1/GPR99, the receptor for 5-oxo-ETE, is associated with a worse prognosis for a GBM patient [9]. OXER1/GPR99 is also a receptor for 2-oxoglutarate, LTC_4_, and LTD_4_ [252]. There is a lack of thorough research on the importance of 5-oxo-ETE in tumorigenesis in GBM tumors.

#### 6.2.2. 12-Lipoxygenase Pathway in Glioblastoma Multiforme

In GBM tumors, expression of 12S-LOX/*ALOX12* and 12R-LOX/*ALOX12B* is not different from healthy brain tissue [8,9], nor is it associated with prognosis severity [9], nor is the expression of the receptor for 12*S*-HETE, i.e., GPR31, elevated and affecting prognosis [8,9]. In contrast, the expression of eLOX3/*ALOXE3* in GBM tumors is lower than in other brain tissue [9,205]. On the other hand, the transcriptomics analysis by Seifert et al. showed no differences between eLOX3/*ALOXE3* expression levels in GBM tumor and healthy brain tissue [8]. Downregulation of eLOX3/*ALOXE3* expression in GBM tumor is associated with increased expression of miR-18a, which downregulates eLOX3/*ALOXE3* expression [205]. At the same time, eLOX3/*ALOXE3* expression is also not related to the prognoses of GBM patients [9].

12-LOX is involved in tumorigenesis in GBM. Studies on various cell lines have shown that 12-LOX expression is common in GBM cancer cells [309]. For this reason, 12-LOX inhibitors inhibit proliferation and reduce the viability of GBM cells [309,322]. 12-LOX inhibitors also inhibit the migration of GBM cells because they reduce the expression of matrix metalloproteinase 2 (MMP2) in these cells [309]. However, the exact mechanism of 12-LOX action on tumorigenic processes in GBM is poorly studied. The fact that eLOX3/*ALOXE3* is anticancer in nature [205] suggests that a lipid mediator not formed by eLOX3/*ALOXE3* is responsible for the pro-cancer properties of 12-LOX. Perhaps it is 12-HETE, a lipid mediator with proven pro-cancer properties in other cancers [323,324]. In addition, higher expression of G2A/GPR132, a receptor for 5-HETE, 12-HETE, 15-HETE, and 9-HODE, is associated with a worse prognosis for a GBM patient (*p* = 0.052) [9]. The oncogenic properties of G2A/GPR132 were also demonstrated in a study on fibroblasts [189], although there is no higher expression of G2A/GPR132 in GBM tumors than in healthy brain tissue [8,9].

12-LOX may also have anti-cancer properties. It converts ARA 20:4n-6 into 12-HpETE, a lipid from the hydroperoxyl group, and for this reason, it can cause lipid peroxidation, which, when free ARA 20:4n-6 is in excess and this PUFA is over-processed, has a destructive effect on the cell [325].

eLOX3/*ALOXE3* has anti-tumor properties in GBM. eLOX3/*ALOXE3* converts 12-HpETE into 12-oxo-ETE. In the absence of eLOX3/*ALOXE3*, 12-HpETE is converted to 12-HETE [205], meaning that eLOX3/*ALOXE3* decreases 12-HETE production. This lipid mediator increases GBM cell migration. When 12-HETE production is decreased, GBM cell migration is reduced.

The lipid mediators produced by eLOX3/*ALOXE3*, including 12-oxo-ETE, have anti-tumor effects, particularly 12-oxo-ETE, which is a ligand for PPARγ [204,205]. Activation of this nuclear receptor inhibits proliferation and induces apoptosis of GBM cancer cells [326,327,328].

The products of eLOX3/*ALOXE3* activity are hepoxilins and trioxilins [200,203], lipid mediators of physiological importance. However, there is a lack of studies on the importance of these lipid mediators in tumorigenesis in GBM.

Analysis on the GEPIA portal [9] and the transcriptomics analysis by Seifert et al. [8] showed no differences in the expression of EPHX2, the enzyme responsible for converting hepoxilins into trioxilins, between GBM tumors and healthy brain tissue [276,280]. At the same time, according to GEPIA, higher EPHX2 expression in GBM tumors is associated with a tendency toward a worse prognosis (*p* = 0.072), which may indicate that hepoxilins and trioxilins may have some role in neoplastic processes in GBM.

#### 6.2.3. 15-Lipoxygenase Pathway in Glioblastoma Multiforme

GEPIA [9] and Seifert et al. [8] showed no differences in the expression of 15-LOX-1/*ALOX15* and 15-LOX-2/*ALOX15B* between GBM tumors and healthy brain tissue. According to GEPIA, the expression level of these enzymes does not affect the prognosis for patients [9]. Studies on various GBM lines have shown differences in the expression of 15-LOX-1/*ALOX15* and 15-LOX-2/*ALOX15B* in GBM cancer cells [309]. 15-LOX is important in the function of GBM cancer cells, and 15-LOX inhibitors reduce the viability and migration of GBM cancer cells [309]. On the other hand, increasing the expression and activity of 15-LOX-1/*ALOX15* throughout the body may have an anti-tumor effect against GBM, as shown by gene therapy using an adenovirus transducing the *ALOX15* gene [329]. This effect may depend on 13-HODE and 15-HETE.

All GBM lineages secrete 13-HODE, a product of the linoleic acid C18:2n-6 conversion with 15-LOX-1/*ALOX15* and 15-LOX-2/*ALOX15B* [266]. 13-HODE increases MMP2 expression in GBM cells, which causes migration [309]. At the same time, 13-HODE also decreases the viability of GBM cells [309], which may depend on the activation of PPARγ via this lipid mediator [290]. This mechanism was confirmed in other cancers, including non-small cell lung cancer [330].

15-HETE can activate G2A/GPR132 [83]. Higher expression of this receptor. according to GEPIA. is associated with a worse prognosis for a GBM patient (*p* = 0.052) [9]. At the same time, the importance of this receptor in GBM has not been thoroughly investigated. Studies in other models have shown that G2A/GPR132 is an oncogene [189]; that is, 15-HETE through activation of G2A/GPR132 has a pro-cancer effect. At the same time, there is no significant upregulation of this receptor expression in GBM tumors [8,9].

The significance of lipoxins in GBM tumors has not been thoroughly investigated. The expression level of the LXA_4_ receptor ALX/FPR2 does not differ between GBM tumors and healthy brain tissue (Table 7) [8,9]. The expression level of this receptor in GBM tumors does not affect prognosis. However, it may be important in tumorigenesis in GBM tumors. Studies on U-87 MG cells have shown that silencing ALX/FPR2 reduces the proliferation and migration of the cells tested [331]. In addition, cells with silenced ALX/FPR2 showed lower expressions of VEGF, a major pro-angiogenic factor. However, this receptor is activated not only by LXA_4_ but also by other factors [332]—for this reason, the importance of LXA_4_ in tumorigenic processes in GBM cannot be determined.

The expression levels of various LOX are not associated with prognoses for GBM patients [9]. This indicates that the LOX pathway is not as relevant to cancer processes as other pathways. For this reason, drugs targeting LOX may show poor efficacy in GBM therapy. At the same time, the analyses performed in this study show that higher expression of OXER1 (the receptor for 5-oxo-ETE) and higher expression of G2A/GPR132 (the receptor for various HETE) are associated with poor prognosis [9]. This indicates a therapeutic target for future drugs developed for the treatment of GBM. In addition, higher expression of GGT1 in GBM tumors is associated with worse prognosis, and higher expression of GGT5 and EPHX2 is associated with a trend of worse prognosis for GBM patients. This indicates a future direction for research into tumor mechanisms in GBM.

### 6.3. Pan-Cancer Analysis of Genes Related to LOX Pathway and GBM

Similar to the COX pathway, we performed a pan-cancer analysis of the expression of the genes involved in the LOX pathway using the data from the GEPIA web server [9].

The expression of eLOX3/*LOXE3* is reduced in GBM tumors. At the same time, there is no change in the expression of this enzyme relative to healthy brain tissue in lower grade gliomas. It is also reduced in two more types of tumors. For this reason, a decrease in eLOX3/*LOXE3* expression may be considered specific to GBM.

In GBM tumors, there is elevated expression of 5-LOX/*ALOX5* and FLAP/*ALOX5AP* relative to healthy brain tissue, which is similar to lower grade gliomas [9]. Expression of these proteins is elevated in 9 and 11 tumor types, respectively. In a similar number of tumor types, there is a reduction in the expressions of 5-LOX/*ALOX5* and FLAP/*ALOX5AP*. This indicates that the elevated expressions of 5-LOX/*ALOX5* and FLAP/*ALOX5AP* may be glioma-specific.

The expression of other LOX is not altered in GBM and lower grade gliomas, which is similar to most other types of cancer. In GBM tumors, there are elevated expressions of LTA_4_H/*LTA4H* and LTC_4_S/*LTC4S* relative to healthy tissue [9]. In lower grade gliomas, there is higher expression of only LTC_4_S/*LTC4S* [9]. According to Seifert et al., in II and III grade gliomas, there are higher expressions of LTA_4_H/*LTA4H* but not LTC_4_S/*LTC4S* relative to healthy brain tissue [8]. LTA_4_H/*LTA4H* expression is elevated in 4 out of 31 analyzed tumor types. LTC_4_S/*LTC4S* is upregulated in six tumor types but downregulated in eleven types [9]. Therefore, the elevated expression of LTA_4_H/*LTA4H* and LTC_4_S/*LTC4S* can be considered as specific to GBM and glioma, respectively.

*GGT5* expression is upregulated in GBM and lower grade gliomas [8,9]. It is downregulated in eleven tumor types and upregulated in seven. Therefore, the elevation of *GGT5* expression can be considered characteristic for gliomas.

*DPEP1* expression is elevated in GBM tumors but not in lower grade gliomas (Table 8) [9]. It is decreased in six types of tumors but increased in four types, including GBM. For this reason, it can be thought that changes in *DPEP1* expression are characteristic of GBM. *EPHX2* expression is often decreased in tumors. In a pan-cancer analysis, 17 types of tumors had a reduced expression of this enzyme relative to healthy tissue. At the same time, in GBM tumors, *EPHX2* expression does not differ relative to healthy brain tissue [8,9].

## 7. Cytochrome P450 Pathway in Glioblastoma Multiforme Tumors

### 7.1. Cytochrome P450 Pathway

In addition to the processing of ARA C20:4n-6 by COX and LOX, this fatty acid can also be converted into lipid mediators with cytochrome P450. It results in the formation of epoxyeicosatrienoic acids (EET) and HETE [333].

ARA C20:4n-6 can undergo either hydroxylation or epoxidation. The ω-hydroxylation reaction converts ARA C20:4n-6 into 20-hydroxyeicosatetraenoic acid (20-HETE). The enzymes responsible for this reaction are CYP1A2 [334], CYP1B1 [335], CYP2U1 [336], CYP4A11 [337,338], CYP4F2 [337,339], CYP4F3A, and CYP4F3B [339].

ARA C20:4n-6 can also be converted to 19-hydroxyeicosatetraenoic acid (19-HETE) in the (ω-1)-hydroxylation reaction. The cytochromes P450 responsible for this are CYP1B1 [335], CYP2C19 [340], CYP2E1 [334], and CYP2U1 [336].

ARA C20:4n-6 can also undergo hydroxylation at other positions with the formation of various HETE [335,340,341,342,343,344]. The cytochromes P450 carrying out this reaction include CYP1A2 [343], CYP1B1 [335], CYP2C9 [334,340], and CYP3A4 [343]. The HETE receptor, with an OH residue at positions 5 to 15, is G2A/GPR132 [83]. In contrast, receptors for 20-HETE include G-protein receptor 75 (GPR75) [345], transient receptor potential vanilloid 1 (TRPV1) channel [346], free fatty acid receptor 1 (FFAR1)/GPR40 [347], and PPARα [348]. HETE can then undergo ω-hydroxylation with CYP4F [333], resulting, for example, in the formation of 10,20-dihydroxyeicosatrienoic acid (10,20-DHET) from 10-HETE, which may be a mechanism for regulating the activity of these lipid mediators.

In the cytochrome P450 pathway, ARA C20:4n-6 can also undergo epoxidation with the formation of epoxyeicosatrienoic acids (EET). Because ARA C20:4n-6 has four double bonds, this reaction produces 5,6-EET, 8,9-EET, 11,12-EET, or 14,15-EET, albeit a given cytochrome P450 can produce mainly only some EET [340]. The enzymes responsible for this reaction are CYP1A2 [334], CYP1B1 [335], CYP2C8 [349,350], CYP2C9 [350], CYP2C19 [340], CYP2J2 [351], and CYP4X1 [352].

The receptor for EET is GPR40 [353]. 14,15-EET can activate receptors for prostaglandins, including PGE_2_ (PTGER_2_, PTGER_3_, and PTGER_4_), PGD_2_ (PTGDR), and PGF_2_α (PTGFR) [354,355]. EET can also activate PPARα (in particular, 11,12-EET [348] and PPARγ [356,357]).

Another important property of EET is that it enters the cell membrane and intracellular membranes. This is as a result of the incorporation of EET into glycerophospholipids at the *sn*-2 position [357,358,359]. EET can also be metabolized by EPHX1 and EPHX2 [357,360]. This is the same enzyme that catalyzes the conversion of hepoxilins (a hydroxy-epoxy derivative of ARA) to trioxilins [276,280]. EET are then converted to dihydroxyeicosatrienoic acid (DHET). In this form, particularly 14,15-DHET, they can activate PPARα [348,361].

EET can also undergo ω-hydroxylation with CYP4F [333]. For example, 8,9-EETs give rise to 20-hydroxy-8(9)-epoxyeicosatrienoic acid (20,8(9)-HEET) (Figure 7) [333,362]. EET can also be converted into either shorter or longer lipid mediators via β-oxidation and elongation, respectively [357]. Another possible reaction is the conversion of 5,6-EET, 8,9-EET, and 11,12-EET with COX [357,363,364], resulting in the formation of lipid mediators with proangiogenic properties. 5,6-epoxy-PGH_2_ is formed from 5,6-EET, [364]. In contrast, 11-hydroxy-8,9-EET (8,9,11-EHET) and 15-hydroxy-8,9-EET (8,9,15-EHET) are formed from 8,9-EET [364,365,366].

HETE and EET are the direct products of cytochrome P450 activity. However, cytochromes p450 are not only involved in the production of these ARA-derived lipid mediators. In addition, CYP4F and CYP4A cause ω-hydroxylation of the already discussed eicosanoids formed in COX and LOX pathways. CYP4A and CYP4F8 are responsible for the ω-hydroxylation and (ω-1)-hydroxylation of prostaglandins, respectively [126,333], and CYP4F is responsible for the transformation of LTB_4_ and lipoxins [333]. The aforementioned reactions often result in the inactivation of these lipid mediators.

It should be mentioned that the aforementioned cytochromes P450 are not only involved in the metabolism of ARA C20:4n-6. They can also metabolize other fatty acids [336], such as linoleic acid [367], and many drugs, including anticancer drugs [349,368].

### 7.2. Cytochrome P450 Pathway in Glioblastoma Multiforme Tumors

ARA C20:4n-6 is converted to 20-HETE [369], which increases the proliferation of GBM cells [370]. 20-HETE may also be an important pro-angiogenic factor in GBM tumors by acting on endothelial cells [369] and enhancing vascular mimicry of GBM cells [371]. Importantly, 20-HETE may not be produced by GBM cells [372] but by TAM and endothelial progenitor cells (EPCs) [373]. CYP2U1 [336,374], whose expression in GBM tumors is elevated relative to healthy brain tissue [8,9], may be responsible for 20-HETE production in GBM tumors. Nevertheless, there is very little research focused on 20-HETE production in GBM tumors.

In the rat glioma RG2 cell line, there is production of various lipid mediators, including 15-HETE, 12-HETE, 8-HETE, 5-HETE, 14,15-diHETE, 14,15-EET, 11,12-diHETE, and 11,12-EET [375]. In part, this may be due to the effect of elevated levels of glutamate in the intercellular space, which is characteristic for GBM tumors [376]. This amino acid increases the expression of CYP1B1 and CYP2U1 in GBM cells [374], leading to increased production of lipid mediators with these cytochrome P450 enzymes.

According to the GEPIA [9] and to Seifert et al. [8], the expression of most of the discussed cytochromes P450 do not differ between GBM tumors and healthy brain tissue. Both sources only show higher expression of *CYP2U1* and lower expression of *CYP4X1* in GBM tumors compared to healthy brain tissue. *CYP2U1* is the cytochrome P450 producing 20-HETE and 19-HETE [336], which shows a possible source of these two lipid mediators in GBM tumors.

GEPIA, in contrast to Seifert et al. shows reduced expression of *CYP2C8* in GBM tumors (Table 9). According to the GEPIA [9], the expression of this cytochrome P450 was not linked to the prognosis of GBM patients. Expression of the receptor for 20-HETE, i.e., *GPR75*, does not differ in GBM tumors compared to healthy brain tissue. The expression level of *GPR75* is not associated with prognosis.

The expression of EPHX1 and EPHX2, enzymes involved in the conversion of EET to DHET, does not differ between GBM tumor and healthy brain tissue [8,9]. In addition, the expression levels of these enzymes are not associated with the prognosis of a GBM patient.

### 7.3. Pan-Cancer Analysis of Cytochrome P450 Genes and Comparison of GBM Expression against Other Cancers

Changes in the expression of various genes in GBM tumors relative to healthy tissue could be the result of tumor-specific neoplastic processes or specific mechanisms found only in GBM. For this reason, a pan-cancer analysis of the expression of the cytochromes P450 genes described above was performed using the GEPIA portal [9].

*CYP2C8* expression was lower in GBM tumors relative to healthy brain tissue [9], similar to lower grade gliomas (Table 10). Downregulation of *CYP2C8* expression occurs in a variety of tumors. Out of 31 analyzed cancers, seven show decreased expression of this enzyme, which shows that reduced expression of *CYP2C8* is common in cancers. In 11 types of cancers out of 31, there is an increase in *CYP2J2* expression. However, in GBM and lower grade gliomas, there is no change in the expression of this cytochrome P450. GEPIA also shows that in 8 out of 31 cancers, including GBM tumors, there is higher expression of *CYP2U1* compared to healthy tissue. This indicates that elevated *CYP2U1* expression may be associated with cancerous processes. In GBM and lower grade gliomas, there is lower expression of *CYP4X1* compared to healthy brain tissue [8,9]. In the other seven types of tumors, there is also a decrease in the expression of this cytochrome p450, which suggests that decreased *CYP4X1* expression in tumor may be a common feature of cancer.

## 8. Conclusions

The importance of the most important ARA C20:4n-6-derived lipid mediators in cancer mechanisms in GBM is very well understood. These compounds, particularly PGE_2_ and leukotrienes, cause the proliferation and migration of GBM cancer cells, are important in the function of GBM cancer stem cells, cause angiogenesis, and by acting on cells of the immune system, inhibit the body’s anti-tumor response. However, the importance in GBM cancer processes of lesser-known ARA C20:4n-6-derived lipid mediators has not yet been investigated. We are talking, for example, about EET, lipoxins, hepoxilins, and some prostanoids, including PGF_2α_ and TxA_2_. Investigating the function of these compounds will provide a better understanding of GBM tumor function. It may also contribute to the development of new therapeutic approaches.

## Figures and Tables

**Figure 1 cancers-15-00946-f001:**
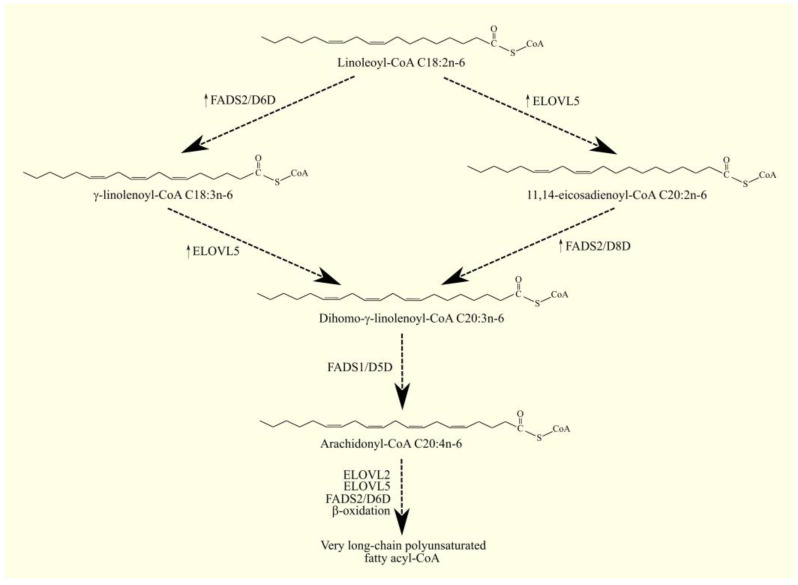
ARA biosynthesis. ARA C20:4n-6 in humans is not synthesized *de novo* but from linoleic acid C18:2n-6. As linoleoyl-CoA C18:2n-6, this PUFA undergoes desaturation to γ-linolenoyl-CoA C18:3n-6 with FADS2/D6D. This fatty acyl-CoA is then converted to dihomo-γ-linolenoyl-CoA C20:3n-6 with ELOVL5 and, finally, to arachidonyl-CoA C20:4n-6 with FADS1/D5D. Dihomo-γ-linolenoyl-CoA C20:3n-6 can also be formed from linoleoyl-CoA via an alternative pathway. Linoleoyl-CoA C18:2n-6 first undergoes elongation with ELOVL5 and then desaturation with FADS2. The latter enzyme in this pathway exhibits Δ^8^-desaturase activity. **↑**—higher expression of given enzymes in GBM tumor relative to healthy tissue.

**Figure 2 cancers-15-00946-f002:**
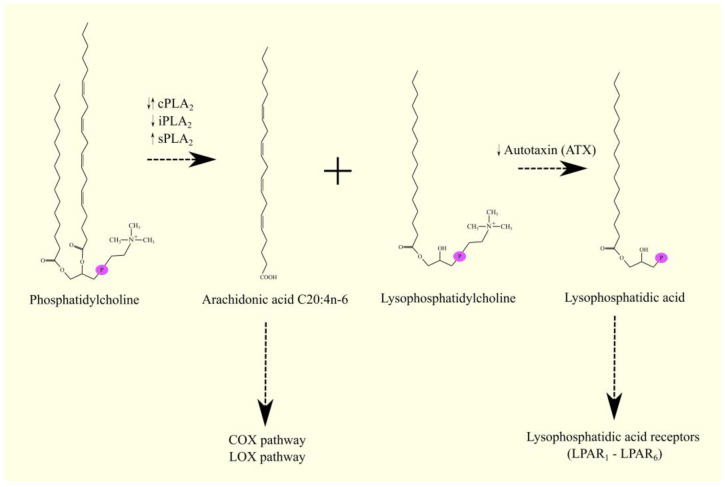
Importance of PLA_2_ in metabolism of ARA and production of lipids mediators from ARA. ARA C20:4n-6 is cleaved from PC by PLA_2_. This reaction also produces LPC, which can be converted in the intercellular space to LPA by ATX. LPA can be considered a lipid mediator because its biological activity is related to the activation of its specific receptors: LPAR_1_-LPAR_6_. Free ARA C20:4n-6, on the other hand, can be used for eicosanoid production in either the COX pathway or the LOX pathway. ↑—higher expression of given enzymes in GBM tumor relative to healthy tissue; ↓—lower expression of given enzymes in GBM tumor relative to healthy tissue.

**Figure 3 cancers-15-00946-f003:**
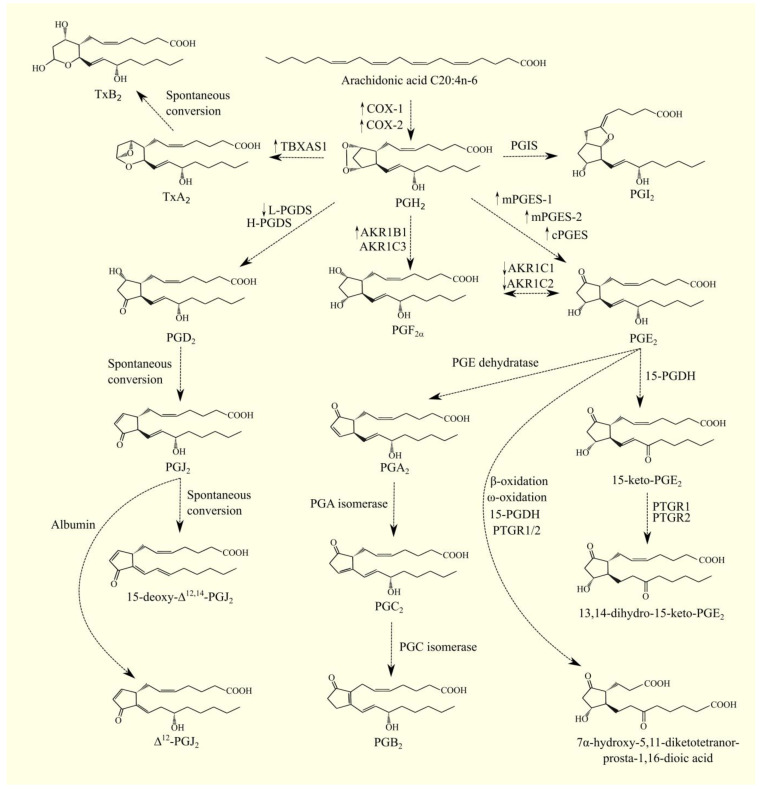
COX pathway. After release by PLA_2_, ARA C20:4n-6 is converted into prostanoids with COX. It is transformed into PGH_2_ with either COX-1 or COX-2. Then, this prostaglandin is transformed into other prostaglandins (PGE_2_, PGD_2_, PGI_2_, and PGF_2α_) or TxA_2_ by the respective synthases. These lipid mediators undergo further transformations. TxA_2_ is unstable and undergoes a spontaneous transformation into TxB_2_. Similarly, PGD_2_ undergoes spontaneous transformation to PGJ_2_—this prostaglandin can then be transformed into 15-deoxy-Δ^12,14^-PGJ_2_ (15d-PGJ_2_) or Δ^12^-PGJ_2_. PGE_2_ can be transformed into PGA_2_, and then into PGC_2_ and PGB_2_. Prostanoids also undergo degradation. The figure shows an example of PGE_2_, which undergoes inactivation by oxidation with 15-PGDH and reduction with PTGR1/2. PGE_2_ can also undergo degradation by β-oxidation and ω-oxidation, followed by the action of 15-PGDH and PTGR1/2. The resulting degradation product is PGE_2_, which is removed from the body. ↑—higher expression of given enzymes in GBM tumor relative to healthy tissue; ↓—lower expression of given enzymes in GBM tumor relative to healthy tissue.

**Figure 4 cancers-15-00946-f004:**
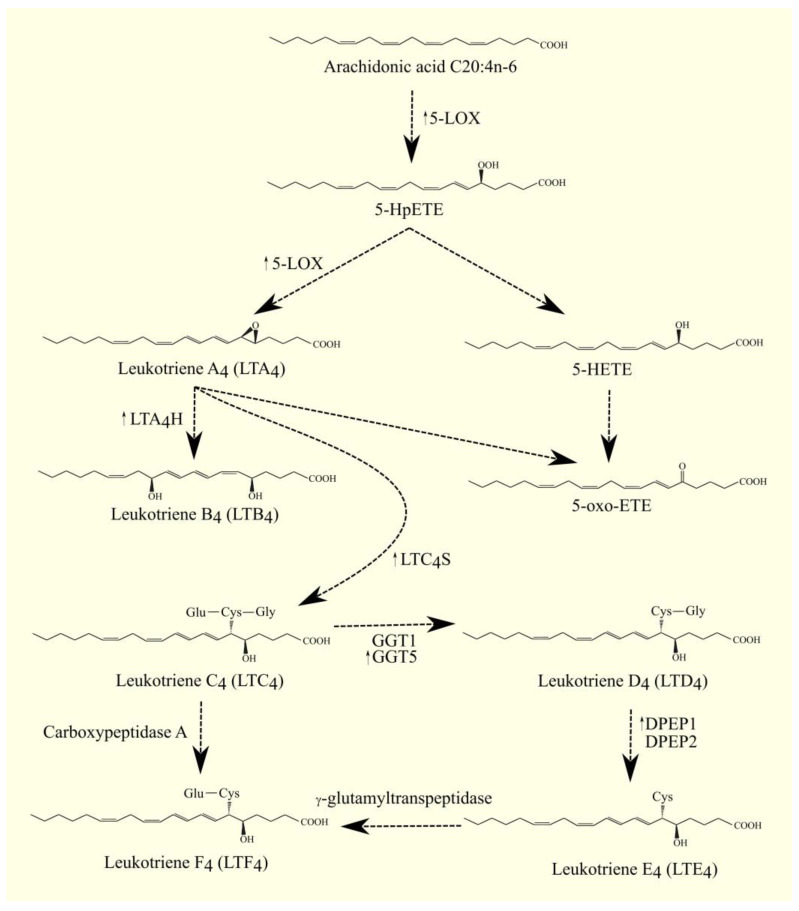
5-LOX pathway. ARA C20:4n-6 is converted to 5-HpETE with 5-LOX. This enzyme also catalyzes the next step in leukotriene biosynthesis. It converts 5-HpETE into LTA_4_, which can then be converted into LTB_4_ with LTA_4_H, into LTC_4_ with LTC_4_S, or into 5-oxo-ETE. 5-HpETE can also be converted to 5-oxo-ETE. LTC_4_ can be converted to other cysteinyl leukotrienes. LTC_4_ can be converted to LTF_4_ with the involvement of carboxypeptidase A or to LTD_4_ with the involvement of GGT1 and GGT5. Subsequently, LTD_4_ can be converted into LTE_4_ with the participation of DPEP1 and DPEP2, and then converted into LTF_4_ with γ-glutamyltranspeptidase. ↑—higher expression of given enzymes in GBM tumor relative to healthy tissue.

**Figure 5 cancers-15-00946-f005:**
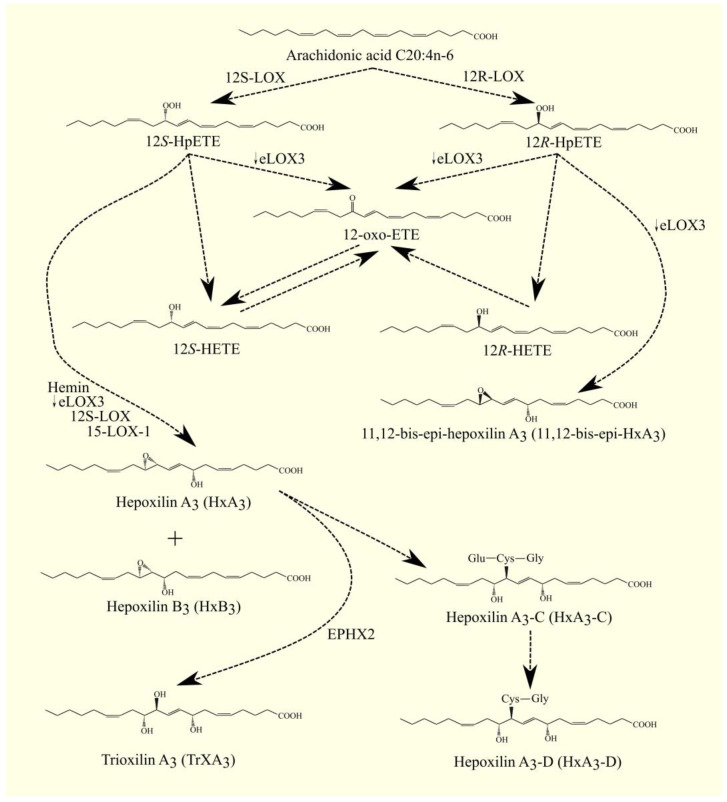
12-LOX pathway. ARA C20:4n-6 is converted to 12*S*-HpETE and 12*R*-HpETE with 12S-LOX and 12R-LOX, respectively. Either 12-oxo-ETE or the corresponding 12-HETE can be formed from these compounds. 12*S*-HpETE can also be converted to HxA_3_ or HxB_3_ with hemin and lipoxygenases: eLOX3, 12S-LOX, or 15-LOX-1. 12*R*-HpETE can undergo a similar conversion to 11,12-bis-epi-HxA_3_. HxA_3_ may undergo further transformations. HxA_3_ can be conjugated to glutathione. HxA_3_-C is then formed, from which amino acids can be detached—HxA_3_-D is then formed in a reaction similar to the transformation of cysteinyl-leukotrienes. HxA_3_ can also be converted to TrXA_3_. Arrows next to enzymes: higher or lower expression of given enzymes in GBM tumor relative to healthy tissue. ↓—lower expression of given enzymes in GBM tumor relative to healthy tissue.

**Figure 6 cancers-15-00946-f006:**
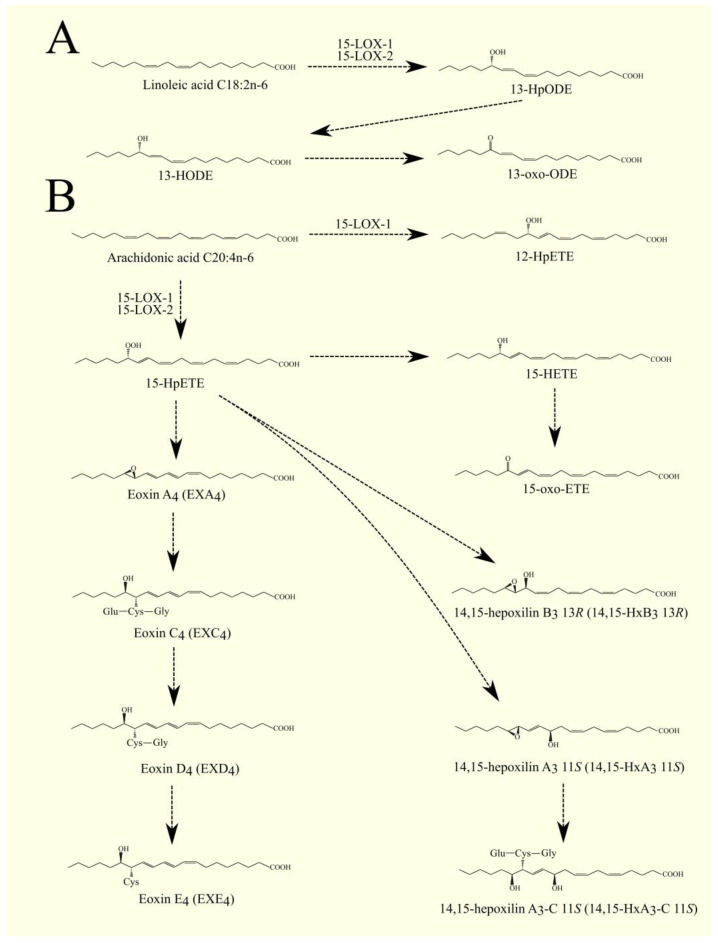
15-LOX pathway. (**A**). Linoleic acid C18:2n-6 can be converted by 15-LOX-1 and 15-LOX-2 into 13-HpODE. This compound can then be converted into 13-HODE and 13-oxo-ODE. (**B**) 15-LOX-1 and 15-LOX-2 can convert ARA C20:4n-6 into 15-HpETE. 15-LOX-1 can also convert this fatty acid into 12-HpETE. 15-HpETE can then be converted into EXA_4_ and into cysteinyl-eoxins EXC_4_, EXD_4_, and EXE_4_. 15-HpETE can also be converted into hepoxilins 14,15-HxA_3_ 11*S*, and 14,15-HxB_3_ 13*R*. 14,15-HxA_3_ 11*S* can be converted to cysteinyl hepoxilins, such as 14,15-HxA_3_-C 11*S*.

**Figure 7 cancers-15-00946-f007:**
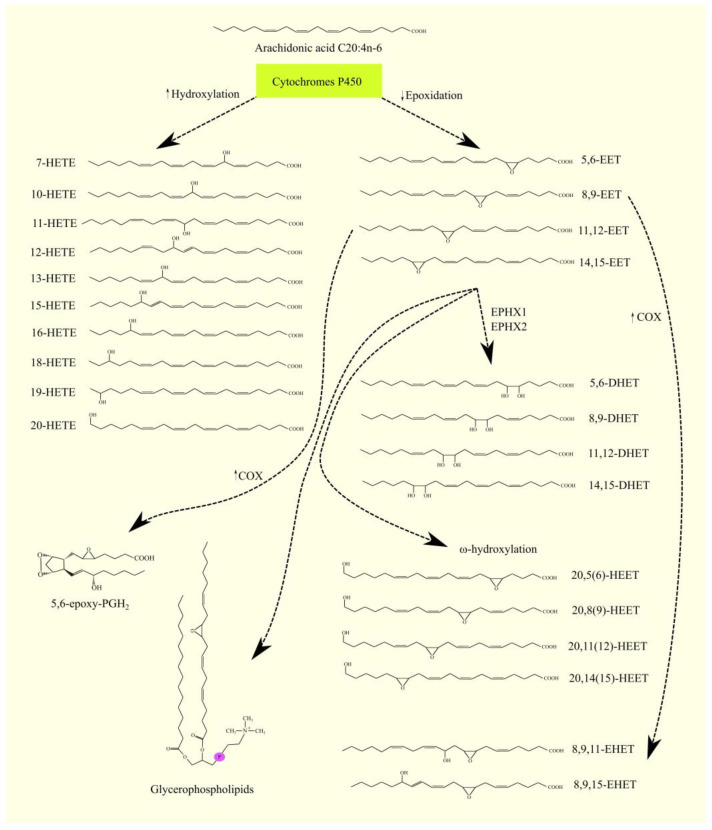
Cytochrome P450 pathway. ARA 20:4n-6 can be converted in the cytochrome P450 pathway, resulting in the formation of various ETT and HETE. ETT can undergo further transformations where they are incorporated into glycerophospholipids in the *sn*-2 position; in this form, they build the cell membrane and intracellular membranes. In addition, the epoxide bond in ETT can be transformed by EPHX1 and EPHX2 into two hydroxyl groups, resulting in the formation of various DHET. ETT can also undergo ω-hydroxylation, which results in the formation of various HEET. ETT can be converted with COX. 5,6-EET then produces 5,6-epoxy-PGH_2_, whereas 8,9-EET produces either 8,9,11-EHET or 8,9,15-EHET. ↑—higher expression of given enzymes in GBM tumor relative to healthy tissue; ↓—lower expression of given enzymes in GBM tumor relative to healthy tissue.

**Table 1 cancers-15-00946-t001:** Description of cPLA_2_ and iPLA_2_.

Name	Expression Level in GBM Tumor Relative to Healthy Tissue		Impact on Prognosis with Higher Expression in GBM Tumors
Source	GEPIA [9]	Seifert et al. [8]	GEPIA [9]
cPLA_2_			
cPLA_2_α/*PLA2G4A*	Higher expression in the tumor	Higher expression in the tumor	No significant impact on prognosis
cPLA_2_β/*PLA2G4B*	Lower expression in the tumor	Expression does not change	No significant impact on prognosis
cPLA_2_γ/*PLA2G4C*	Expression does not change	Lower expression in the tumor	No significant impact on prognosis
cPLA_2_δ/*PLA2G4D*	Expression does not change	Expression does not change	No significant impact on prognosis
cPLA_2_ε/*PLA2G4E*	Expression does not change	Expression does not change	No significant impact on prognosis
cPLA_2_ζ/*PLA2G4F*	Expression does not change	Expression does not change	No significant impact on prognosis
iPLA_2_			
iPLA_2_β/*PLA2G6*	Expression does not change	Lower expression in the tumor	No significant impact on prognosis
iPLA_2_γ/*PNPLA8*	Expression does not change	Expression does not change	No significant impact on prognosis
iPLA_2_δ/*PNPLA6*	Expression does not change	Lower expression in the tumor	No significant impact on prognosis
iPLA_2_ε/*PNPLA3*	Expression does not change	Expression does not change	No significant impact on prognosis
iPLA_2_ζ/*PNPLA2*	Expression does not change	Expression does not change	Worse prognosis *p* = 0.087
iPLA_2_η/*PNPLA4*	Expression does not change	Expression does not change	Worse prognosis

Red background—higher expression in the tumor; blue background—lower expression in the tumor; red background—worse prognosis with higher expression of a given PLA_2_.

**Table 2 cancers-15-00946-t002:** Description of sPLA_2_ and sPLA_2_ receptors in GBM.

Name	Expression Level in GBM Tumor Relative to Healthy Tissue		Impact on Prognosis with Higher Expression in GBM Tumors	
Source	GEPIA [9]	Seifert et al. [8]	GEPIA [9]	Wu et al. [52]
*PLA2G1B*	Expression does not change	Expression does not change	Worse prognosis *p* = 0.078	Worse prognosis
*PLA2G2A*	Higher expression in the tumor	Higher expression in the tumor	No significant impact on prognosis	No significant impact on prognosis
*PLA2G2D*	Expression does not change	Expression does not change	No significant impact on prognosis	No significant impact on prognosis
*PLA2G2E*	Expression does not change	Expression does not change	N/A	Worse prognosis
*PLA2G2F*	Expression does not change	Expression does not change	N/A	No significant impact on prognosis
*PLA2G3*	Expression does not change	Expression does not change	No significant impact on prognosis	Worse prognosis
*PLA2G5*	Higher expression in the tumor	Higher expression in the tumor	No significant impact on prognosis	Worse prognosis
*PLA2G7*	Expression does not change	Expression does not change	No significant impact on prognosis	No significant impact on prognosis
*PLA2G10*	Expression does not change	Expression does not change	N/A	No significant impact on prognosis
*PLA2G12A*	Higher expression in the tumor	Expression does not change	No significant impact on prognosis	No significant impact on prognosis
*PLA2G12B*	Expression does not change	Expression does not change	N/A	No significant impact on prognosis
*PLA2G15*	Higher expression in the tumor	Expression does not change	Worse prognosis	No significant impact on prognosis
*PLA2G16*	Expression does not change	Expression does not change	No significant impact on prognosis	No significant impact on prognosis
PLA_2_R1/*PLA2R1*	Expression does not change	Expression does not change	Worse prognosis	

Red background—higher expression in the tumor; red background—worse prognosis with higher expression of a given PLA_2_.

**Table 3 cancers-15-00946-t003:** Pan-cancer analysis of gene expression of cPLA_2_ and iPLA_2_.

Name of Cancer	cPLA_2_α/*PLA2G4A*	cPLA_2_β/*PLA2G4B*	cPLA_2_γ/*PLA2G4C*	cPLA_2_δ/*PLA2G4D*	cPLA_2_ε/*PLA2G4E*	cPLA_2_ζ/*PLA2G4F*	iPLA_2_β/*PLA2G6*	iPLA_2_γ/*PNPLA8*	iPLA_2_δ/*PNPLA6*	iPLA_2_ε/*PNPLA3*	iPLA_2_ζ/*PNPLA2*	iPLA_2_η/*PNPLA4*
Adrenocortical carcinoma (ACC)	↓	↓	=	=	=	=	↓	=	=	=	=	=
Bladder urothelial carcinoma (BLCA)	↓	=	↓	=	=	=	=	=	=	=	=	=
Breast invasive carcinoma (BRCA)	↓	↓	=	=	=	=	↓	=	=	=	↓	=
Cervical squamous cell carcinoma and endocervical adenocarcinoma (CESC)	=	↓	↓	=	=	↑	↓	=	=	=	=	=
Cholangiocarcinoma (CHOL)	=	=	↑	=	=	↑	↑	=	↑	↓	↑	=
Colon adenocarcinoma (COAD)	↓	↓	↓	=	=	↑	↓	=	=	=	=	↑
Lymphoid neoplasm diffuse large B-cell lymphoma (DLBC)	=	=	↑	=	=	=	=	=	=	=	↓	↑
Esophageal carcinoma (ESCA)	=	↓	=	=	=	↓	=	=	=	=	=	=
Glioblastoma multiforme (GBM)	↑	↓	=	=	=	=	=	=	=	=	=	=
Head and neck squamous cell carcinoma (HNSC)	=	↓	=	=	=	=	=	=	=	=	=	=
Kidney chromophobe (KICH)	=	↓	=	=	=	↑	=	=	=	=	=	=
Kidney renal clear cell carcinoma (KIRC)	↓	=	=	=	=	↓	=	=	=	=	=	=
Kidney renal papillary cell carcinoma (KIRP)	=	=	=	=	=	↓	=	=	=	=	=	=
Acute myeloid leukemia (LAML)	↑	↑	↑	=	=	=	↑	=	↑	=	↑	↓
Brain lower grade glioma (LGG)	=	↓	=	=	=	=	=	↑	=	=	=	=
Liver hepatocellular carcinoma (LIHC)	=	↓	↑	=	=	=	=	=	=	=	=	=
Lung adenocarcinoma (LUAD)	↑	↓	↓	=	=	↓	↓	=	↓	=	↓	=
Lung squamous cell carcinoma (LUSC)	=	↓	↓	=	=	↓	↓	=	↓	=	↓	↑
Ovarian serous cystadenocarcinoma (OV)	=	↓	=	=	=	=	↓	=	=	=	=	=
Pancreatic adenocarcinoma (PAAD)	↑	=	↑	=	=	=	=	↑	↑	=	=	=
Pheochromocytoma and paraganglioma (PCPG)	↓	=	=	=	=	=	=	=	=	=	↓	=
Prostate adenocarcinoma (PRAD)	=	↓	=	=	=	=	↓	=	=	=	=	=
Rectum adenocarcinoma (READ)	↓	↓	↓	=	=	↑	↓	=	=	=	=	↑
Sarcoma (SARC)	=	=	=	=	=	↓	=	=	=	=	=	=
Skin cutaneous melanoma (SKCM)	=	↓	↑	↓	↓	↓	↓	=	=	↓	↓	=
Stomach adenocarcinoma (STAD)	↑	↓	=	=	=	=	↓	=	=	=	=	=
Testicular germ cell tumors (TGCT)	=	↓	↓	=	=	↑	↓	=	↓	=	↓	↑
Thyroid carcinoma (THCA)	=	↓	=	=	=	↓	↓	=	=	=	↓	=
Thymoma (THYM)	=	=	↑	=	=	↑	↑	↑	=	=	=	↑
Uterine corpus endometrial carcinoma (UCEC)	=	↓	↓	=	=	=	↓	=	=	=	=	=
Uterine carcinosarcoma (UCS)	↓	↓	↓	=	=	=	↓	=	=	=	↓	=

Red background, ↑—expression higher in tumors than in healthy tissue; blue background, ↓—expression lower in tumors than in healthy tissue; gray background, =—expression does not differ between tumors and healthy tissue.

**Table 4 cancers-15-00946-t004:** Pan-cancer analysis of gene expression of sPLA_2_ and sPLA_2_ receptors.

Name of Cancer	*PLA2G1B*	*PLA2G2A*	*PLA2G2D*	*PLA2G2E*	*PLA2G2F*	*PLA2G3*	*PLA2G5*	*PLA2G7*	*PLA2G10*	*PLA2G12A*	*PLA2G12B*	*PLA2G15*	*PLA2G16*	*PLA2R1*
Adrenocortical carcinoma (ACC)	↑	↓	=	=	=	=	=	=	=	=	=	=	=	=
Bladder urothelial carcinoma (BLCA)	=	↓	=	=	↑	=	↓	↑	=	=	=	=	=	=
Breast invasive carcinoma (BRCA)	=	↓	=	=	=	=	↓	=	=	=	=	=	↓	↓
Cervical squamous cell carcinoma and endocervical adenocarcinoma (CESC)	↓	↓	=	=	=	=	=	↑	=	=	=	=	=	↓
Cholangiocarcinoma (CHOL)	=	↓	=	=	=	=	=	=	=	=	↓	=	=	=
Colon adenocarcinoma (COAD)	=	=	=	=	=	=	↓	↑	↑	=	↑	=	=	=
Lymphoid neoplasm diffuse large B-cell lymphoma (DLBC)	=	=	↑	=	=	=	=	↑	=	↑	=	↑	↑	=
Esophageal carcinoma (ESCA)	↓	↓	=	=	=	↓	↓	↑	↑	=	=	=	↑	=
Glioblastoma multiforme (GBM)	=	↑	=	=	=	=	↑	=	=	↑	=	↑	=	=
Head and neck squamous cell carcinoma (HNSC)	=	↓	=	=	=	=	=	↑	=	=	=	=	↓	=
Kidney chromophobe (KICH)	=	=	=	=	=	=	=	=	=	=	=	=	=	↓
Kidney renal clear cell carcinoma (KIRC)	=	=	=	=	=	=	=	↑	=	=	=	=	↑	↓
Kidney renal papillary cell carcinoma (KIRP)	=	=	↑	=	=	=	=	↑	=	=	↓	=	↑	↓
Acute myeloid leukemia (LAML)	=	=	=	=	=	↓	=	=	=	↓	=	=	=	=
Brain lower grade glioma (LGG)	=	=	=	=	=	=	=	=	=	=	=	=	=	=
Liver hepatocellular carcinoma (LIHC)	=	↓	=	=	=	=	=	=	=	=	=	=	=	=
Lung adenocarcinoma (LUAD)	↓	↓	↑	=	=	↓	↓	=	=	=	=	=	=	=
Lung squamous cell carcinoma (LUSC)	↓	↓	↑	=	=	=	↓	=	↓	=	=	=	↓	=
Ovarian serous cystadenocarcinoma (OV)	↓	↓	=	=	=	=	↓	=	=	=	=	=	↓	↓
Pancreatic adenocarcinoma (PAAD)	↓	↓	=	=	=	=	↑	↑	↑	=	=	↑	↑	↑
Pheochromocytoma and paraganglioma (PCPG)	↓	↓	=	=	=	=	=	=	=	=	=	=	=	=
Prostate adenocarcinoma (PRAD)	=	↑	=	=	=	=	=	↑	=	↑	=	=	=	=
Rectum adenocarcinoma (READ)	=	=	=	=	=	=	↓	↑	↑	=	↑	=	=	=
Sarcoma (SARC)	=	=	=	=	=	=	=	=	=	=	↓	=	=	↓
Skin cutaneous melanoma (SKCM)	=	↓	↑	=	↓	↓	=	↑	=	=	=	=	↑	↓
Stomach adenocarcinoma (STAD)	↓	↑	=	=	=	=	=	↑	↑	=	=	=	=	=
Testicular germ cell tumors (TGCT)	↓	↓	↑	=	=	=	↓	↑	↓	=	=	=	↓	=
Thyroid carcinoma (THCA)	=	↓	=	=	=	=	=	=	=	=	↑	=	↑	↓
Thymoma (THYM)	=	=	=	=	=	=	=	↑	=	↑	=	↑	↑	↑
Uterine corpus endometrial carcinoma (UCEC)	↓	↓	=	=	=	=	↓	↑	↑	↑	=	=	=	↓
Uterine carcinosarcoma (UCS)	↓	↓	=	=	=	=	↓	↑	=	=	=	=	=	↓

Red background, ↑—expression higher in tumor than in healthy tissue; blue background, ↓—expression lower in tumor than in healthy tissue; gray background, =—expression does not differ between tumor and healthy tissue.

**Table 5 cancers-15-00946-t005:** Description of individual enzymes involved in the synthesis, transport, and degradation of prostaglandins.

Name	Biochemical Significance	Expression Level in GBM Tumor Relative to Healthy Tissue			Impact on Prognosis with Higher Expression in GBM Tumors	
Source		GEPIA [9]	Seifert et al. [8]	Other Data Source	GEPIA [9]	Other Data Source
COX-1	PGH_2_ synthesis from ARA	Higher expression in the tumor	Higher expression in the tumor	Higher expression in the tumor [141]	No significant impact on prognosis	No significant impact on prognosis [121]
COX-2	PGH_2_ synthesis from ARA	Expression does not change	Expression does not change	Higher expression in the tumor [141,142]	No significant impact on prognosis	Worse prognosis [160,187,188]
mPGES-1	PGE_2_ synthesis from PGH_2_	Expression does not change	Expression does not change	Higher expression in the tumor [143]	Worse prognosis	Worse prognosis [121]
mPGES-2	PGE_2_ synthesis from PGH_2_	Expression does not change	Expression does not change	Higher expression in the tumor [143]	No significant impact on prognosis	No significant impact on prognosis [121]
cPGES	PGE_2_ synthesis from PGH_2_	Higher expression in the tumor	Expression does not change	Higher expression in the tumor [143]	No significant impact on prognosis	
H-PGDS	Synthesis of PGD_2_ from PGH_2_	Higher expression in the tumor	Lower expression in the tumor		No significant impact on prognosis	
L-PGDS	Synthesis of PGD_2_ from PGH_2_	Lower expression in the tumor	Lower expression in the tumor		No significant impact on prognosis	
TBXAS1	TxA_2_ synthesis from PGH_2_	Higher expression in the tumor	Higher expression in the tumor	Higher expression in the tumor [144]	No significant impact on prognosis	
AKR1B1	PGF_2α_ synthesis from PGH_2_	Higher expression in the tumor	Higher expression in the tumor		Worse prognosis	
AKR1C1	PGF_2α_ synthesis from PGE_2_	Lower expression in the tumor	Lower expression in the tumor		No significant impact on prognosis	
AKR1C2	PGF_2α_ synthesis from PGE_2_	Lower expression in the tumor	Expression does not change		No significant impact on prognosis	
AKR1C3	PGF_2α_ synthesis from PGH_2_	Expression does not change	Expression does not change		No significant impact on prognosis	
PGIS	PGIF_2_ synthesis from PGH_2_	Expression does not change	Expression does not change		No significant impact on prognosis	
MRP4	Secretion of prostaglandins from the cell	Higher expression in the tumor	Expression does not change		No significant impact on prognosis	
PGT/SLCO2A1	Uptake of prostaglandins into the cell	Expression does not change	Expression does not change		No significant impact on prognosis	
15-PGDH	First degradation reaction/formation of PPARγ ligand from PGE_2_	Expression does not change	Expression does not change		No significant impact on prognosis	Better prognosis [121]
PTGR1	Second degradation/inactivation reaction of PPARγ ligand made from PGE_2_	Expression does not change	Expression does not change		No significant impact on prognosis	Worse prognosis [121]
PTGR2	Second degradation/inactivation reaction of PPARγ ligand made from PGE_2_	Expression does not change	Expression does not change		No significant impact on prognosis	No significant impact on prognosis

Red background—higher expression in the tumor; blue background—lower expression in the tumor; red background—worse prognosis with higher expression; blue background—better prognosis with higher expression.

**Table 6 cancers-15-00946-t006:** Pan-cancer analysis of expression of genes involved in COX pathway.

Name of Cancer	COX-1/*PTGS1*	COX-2/*PTGS2*	mPGES-1/*PTGES*	mPGES-2/*PTGES2*	cPGES/*PTGES3*	*H-PGDS/HPGDS*	*L-PGDS/PTGDS*	*TBXAS1*	*AKR1B1*	*AKR1C1*	*AKR1C2*	*AKR1C3*	PGIS/*PTGIS*	MRP4/*ABCC4*	PGT/*SLCO2A1*	15-PGDH/*HPGD*	*PTGR1*	*PTGR2*
Adrenocortical carcinoma (ACC)	=	=	=	=	↑	=	↓	↓	↓	=	↓	=	↓	=	↓	↑	↓	=
Bladder urothelial carcinoma (BLCA)	↓	↓	=	=	=	↓	↓	=	=	↓	=	=	↓	↓	↓	↓	=	=
Breast invasive carcinoma (BRCA)	=	↓	=	=	=	=	↓	=	=	↓	↓	↓	↓	=	=	↓	=	=
Cervical squamous cell carcinoma and endocervical adenocarcinoma (CESC)	=	=	=	=	=	=	↓	=	=	=	=	=	↓	=	↓	↓	=	=
Cholangiocarcinoma (CHOL)	↑	=	↑	↑	↑	=	=	=	↑	=	=	↑	=	↑	↑	↓	↓	=
Colon adenocarcinoma (COAD)	↓	=	=	=	↑	↓	↓	↑	↓	↓	↓	=	↓	=	↓	↓	=	=
Lymphoid neoplasm diffuse large B-cell lymphoma (DLBC)	=	↓	↑	↑	↑	=	↑	↓	↑	=	=	=	=	=	↑	↓	↑	↑
Esophageal carcinoma (ESCA)	↓	↑	=	=	=	=	↓	=	=	↓	↓	↑	↓	↑	=	↓	↓	=
Glioblastoma multiforme (GBM)	↑	=	=	=	↑	↑	↓	↑	↑	↓	↓	=	=	↑	=	=	=	=
Head and neck squamous cell carcinoma (HNSC)	=	=	=	=	=	=	↓	=	↑	=	=	=	=	=	=	↓	=	=
Kidney chromophobe (KICH)	=	=	=	=	=	=	↓	=	=	=	↓	↓	↓	=	↓	=	↓	=
Kidney renal clear cell carcinoma (KIRC)	↑	=	↓	=	=	=	↓	↑	=	=	=	=	=	=	=	↓	↓	=
Kidney renal papillary cell carcinoma (KIRP)	=	↓	=	=	=	=	↓	=	↑	↑	↑	=	↓	=	↓	↓	=	=
Acute myeloid leukemia (LAML)	=	↑	=	↓	↓	↑	↓	↑	↑	↓	↓	↑	=	↓	=	=	↑	↓
Brain lower grade glioma (LGG)	↑	=	=	=	=	↑	=	↑	=	=	=	↑	=	↑	=	=	=	↑
Liver hepatocellular carcinoma (LIHC)	=	=	=	=	↑	=	=	=	=	↑	↑	↑	↓	=	=	↓	=	=
Lung adenocarcinoma (LUAD)	=	=	↑	=	=	=	↓	=	=	=	=	=	↓	=	↓	↓	=	=
Lung squamous cell carcinoma (LUSC)	=	=	↑	=	=	↓	↓	↓	=	↑	↑	↑	↓	=	↓	↓	=	=
Ovarian serous cystadenocarcinoma (OV)	↑	=	=	=	=	=	↑	=	=	↓	↓	↓	↓	=	=	↑	=	=
Pancreatic adenocarcinoma (PAAD)	↑	↑	↑	↑	↑	↑	↑	↑	↑	↑	↑	↑	↑	↑	↑	↑	↑	=
Pheochromocytoma and paraganglioma (PCPG)	=	=	↑	=	=	=	↓	=	↓	=	=	=	=	=	=	=	↓	=
Prostate adenocarcinoma (PRAD)	=	↓	=	=	=	=	↓	=	↓	↓	↓	=	↓	↑	↓	=	=	=
Rectum adenocarcinoma (READ)	↓	↓	↑	=	↑	=	↓	↑	↓	↓	↓	↑	↓	=	↓	↓	=	=
Sarcoma (SARC)	↓	=	=	=	=	=	=	=	=	=	=	=	=	=	=	↓	=	=
Skin cutaneous melanoma (SKCM)	↓	=	↓	=	=	↓	=	↑	↑	↓	↓	↓	=	=	↓	↓	=	=
Stomach adenocarcinoma (STAD)	=	=	=	=	↑	=	=	=	=	↓	↓	=	=	=	=	=	=	=
Testicular germ cell tumors (TGCT)	=	=	↓	=	↑	=	↓	↑	=	↓	↓	↓	=	=	=	=	↑	↓
Thyroid carcinoma (THCA)	=	=	=	=	=	=	↓	=	=	↓	↓	↓	↓	=	↓	=	=	=
Thymoma (THYM)	↓	↓	=	↑	↑	↑	↑	↓	↑	=	↑	↑	=	=	↑	=	↑	↑
Uterine corpus endometrial carcinoma (UCEC)	↑	=	=	=	=	↓	↓	=	=	↓	=	=	↓	=	↓	↓	=	=
Uterine carcinosarcoma (UCS)	=	=	↑	=	=	=	↓	=	=	=	=	=	↓	=	↓	↓	=	=

Red background, ↑—expression higher in tumor than in healthy tissue; blue background, ↓—expression lower in tumor than in healthy tissue; gray background, =—expression does not differ between tumor and healthy tissue.

**Table 7 cancers-15-00946-t007:** Description of the various enzymes involved in the synthesis, action, and degradation of lipoxygenases along with their involvement in tumorigenesis in GBM.

Name	Biochemical Significance	Expression Level In GBM Tumors Relative To Healthy Tissue		Impact on Prognosis with Higher Expression in GBM Tumors
Source		GEPIA [9]	Seifert et al. [8]	GEPIA [9]
eLOX3/*ALOXE3*	Production of hepoxilins/hydroxy-epoxyeicosatrienoic acid and oxo-ETE from HpETE	Lower expression in the tumor	Expression does not change	No significant impact on prognosis
5-LOX/ALOX5	5-HpETE production from ARA; the first enzyme in leukotrienes and the 5-oxo-ETE synthesis pathway; synthesis of lipoxins from 15-HpETE and 15-HETE	Higher expression in the tumor	Higher expression in the tumor	No significant impact on prognosis
FLAP/ALOX5AP	Substrate carrier for 5-LOX	Higher expression in the tumor	Higher expression in the tumor	No significant impact on prognosis
12S-LOX/ALOX12	12*S*-HpETE production from ARA; the first enzyme in the hepoxilin production pathway; production of lipoxins from LTA_4_	Expression does not change	Expression does not change	No significant impact on prognosis
12R-LOX/ALOX12B	12*R*-HpETE production from ARA	Expression does not change	Expression does not change	No significant impact on prognosis
15-LOX-1/ALOX15	15-HpETE production from ARA; 12-HpETE production from ARA; production of lipoxins, eoxins, 15-oxo-ETE and 15-HETE; production of 13-HpODE from linoleic acid C18:2n-6	Expression does not change	Expression does not change	No significant impact on prognosis
15-LOX-2/ALOX15B	15-HpETE production from ARA; production of 15-HpETE, lipoxins, eoxins, 15-oxo-ETE and 15-HETE	Expression does not change	Expression does not change	No significant impact on prognosis
LTA_4_H	LTB_4_ production from LTA_4_	Higher expression in the tumor	Higher expression in the tumor	No significant impact on prognosis
LTC_4_S	LTC_4_ production from LTA_4_	Higher expression in the tumor	Expression does not change	No significant impact on prognosis
GGT1	LTD_4_ production from LTC_4_	Expression does not change	Expression does not change	Worse prognosis
GGT5	LTD_4_ production from LTC_4_	Higher expression in the tumor	Higher expression in the tumor	Worse prognosis (*p* = 0.055)
DPEP1	LTE_4_ production from LTD_4_	Higher expression in the tumor	Expression does not change	No significant impact on prognosis
DPEP2	LTE_4_ production from LTD_4_	Expression does not change	Expression does not change	No significant impact on prognosis
EPHX2	Conversion of hepoxilins into trioxilin	Expression does not change	Expression does not change	Worse prognosis (*p* = 0.072)
Receptors				
LTB_4_R1	LTB_4_ receptor	Expression does not change	Expression does not change	No significant impact on prognosis
LTB_4_R2	LTB_4_ receptor	Expression does not change	Expression does not change	No significant impact on prognosis
CYSLTR_1_	Cysteinyl-leukotrienes receptor	Expression does not change	Expression does not change	No significant impact on prognosis
CYSLTR_2_	Cysteinyl-leukotrienes receptor	Expression does not change	Expression does not change	No significant impact on prognosis
OXER1	5-oxo-ETE receptor	Expression does not change	Expression does not change	Worse prognosis
ALX/FPR2	LXA_4_ receptor	Expression does not change	Expression does not change	No significant impact on prognosis
GPR17	Cysteinyl-leukotrienes receptor	Expression does not change	Expression does not change	No significant impact on prognosis
GPR31	12*S*-HETE receptor	Expression does not change	Expression does not change	No significant impact on prognosis
OXGR1/GPR99	LTE_4_ receptor	Expression does not change	Expression does not change	No significant impact on prognosis
G2A/GPR132	5-HETE, 12-HETE, 15-HETE, 9-HODE receptor	Expression does not change	Expression does not change	Worse prognosis (*p* = 0.052)

Red background—higher expression in the tumor; blue background—lower expression in the tumor; red background—worse prognosis with higher expression.

**Table 8 cancers-15-00946-t008:** Pan-cancer analysis of expression of genes involved in the LOX pathway.

Name of Cancer	eLOX3/*ALOXE3*	5-LOX/*ALOX5*	FLAP/*ALOX5AP*	12S-LOX/*ALOX12*	12R-LOX/*ALOX12B*	15-LOX-1/*ALOX15*	15-LOX-2/*ALOX15B*	LTA_4_H/*LTA4H*	LTC_4_S/*LTC4S*	*GGT1*	*GGT5*	*DPEP1*	*DPEP2*	*EPHX2*
Adrenocortical carcinoma (ACC)	=	↓	↓	=	=	=	↓	=	=	=	↓	=	=	↓
Bladder urothelial carcinoma (BLCA)	=	=	↓	=	=	=	=	=	↓	=	↓	=	=	↓
Breast invasive carcinoma (BRCA)	=	=	=	=	=	=	↓	=	=	=	=	=	=	=
Cervical squamous cell carcinoma and endocervical adenocarcinoma (CESC)	=	=	=	=	=	=	=	=	↓	=	↓	=	=	↓
Cholangiocarcinoma (CHOL)	=	↑	↑	=	=	=	=	↑	↑	=	=	=	=	↓
Colon adenocarcinoma (COAD)	=	=	=	=	=	=	=	=	↓	↑	=	↑	↓	=
Lymphoid neoplasm diffuse large B-cell lymphoma (DLBC)	=	↓	↓	↓	=	=	↑	=	=	=	↑	=	↓	=
Esophageal carcinoma (ESCA)	=	=	=	↓	=	=	↓	=	=	=	=	=	=	↓
Glioblastoma multiforme (GBM)	↓	↑	↑	=	=	=	=	↑	↑	=	↑	↑	=	=
Head and neck squamous cell carcinoma (HNSC)	↑	=	=	↓	=	=	=	=	=	=	↑	=	=	↓
Kidney chromophobe (KICH)	=	=	=	=	=	=	=	=	↓	↓	↓	↓	=	=
Kidney renal clear cell carcinoma (KIRC)	=	↑	↑	=	=	=	=	=	=	↑	=	↓	↑	↓
Kidney renal papillary cell carcinoma (KIRP)	=	↑	↑	=	=	=	↑	=	=	↑	↓	↓	↑	=
Acute myeloid leukemia (LAML)	=	↑	↑	=	=	=	=	↑	↑	=	↑	=	↑	↓
Brain lower grade glioma (LGG)	=	↑	↑	=	=	=	=	=	↑	=	↑	=	=	=
Liver hepatocellular carcinoma (LIHC)	=	=	=	=	=	=	=	=	=	=	↓	=	=	↓
Lung adenocarcinoma (LUAD)	=	↓	↓	=	=	=	↓	=	↓	=	=	=	↓	=
Lung squamous cell carcinoma (LUSC)	=	↓	↓	=	=	=	↓	↓	↓	↓	=	=	↓	=
Ovarian serous cystadenocarcinoma (OV)	=	↑	↑	=	=	=	=	=	=	=	↓	=	=	↓
Pancreatic adenocarcinoma (PAAD)	=	↑	↑	=	=	=	↑	↑	↑	↑	↑	↓	↑	↓
Pheochromocytoma and paraganglioma (PCPG)	=	=	↓	=	=	=	=	=	=	=	↓	=	=	↓
Prostate adenocarcinoma (PRAD)	=	=	=	=	=	=	=	=	=	↑	=	=	=	=
Rectum adenocarcinoma (READ)	=	=	=	=	=	=	=	=	↓	↑	=	↑	=	=
Sarcoma (SARC)	=	=	=	=	=	=	=	=	=	↓	=	↓	=	↓
Skin cutaneous melanoma (SKCM)	↓	↓	=	↓	↓	=	↓	=	↓	=	↓	=	=	↓
Stomach adenocarcinoma (STAD)	=	=	↑	=	=	=	=	=	=	=	=	=	=	=
Testicular germ cell tumors (TGCT)	↓	=	↑	=	=	↓	=	=	↓	=	=	↓	=	↓
Thyroid carcinoma (THCA)	=	↑	↑	=	=	=	↑	=	=	=	=	=	=	=
Thymoma (THYM)	=	↓	↓	=	=	=	=	=	↑	=	↑	↑	↓	↑
Uterine corpus endometrial carcinoma (UCEC)	=	↓	=	=	=	=	=	=	↓	↑	↓	=	=	↓
Uterine carcinosarcoma (UCS)	=	↓	↓	=	=	=	=	=	↓	=	↓	=	=	↓

Red background, ↑—expression higher in tumor than in healthy tissue; blue background, ↓—expression lower in tumor than in healthy tissue; gray background, =—expression does not differ between tumor and healthy tissue.

**Table 9 cancers-15-00946-t009:** Significance of cytochromes P450 and GPR75 receptors in ARA metabolism and tumorigenic processes in GBM.

Name	Expression Level in GBM Tumor Relative to Healthy Tissue		Impact on Prognosis with Higher Expression in GBM Tumors
Source	GEPIA [9]	Seifert et al. [8]	GEPIA [9]
*CYP1A2*	Expression does not change	Expression does not change	N/A
*CYP1B1*	Expression does not change	Expression does not change	No significant impact on prognosis
*CYP2C8*	Lower expression in the tumor	Expression does not change	No significant impact on prognosis
*CYP2C9*	Expression does not change	Expression does not change	N/A
*CYP2C19*	Expression does not change	Expression does not change	N/A
*CYP2J2*	Expression does not change	Expression does not change	No significant impact on prognosis
*CYP2U1*	Higher expression in the tumor	Higher expression in the tumor	No significant impact on prognosis
*CYP3A4*	Expression does not change	Expression does not change	Worse prognosis *p* = 0.07
*CYP4A11*	Expression does not change	Expression does not change	No significant impact on prognosis
*CYP4F2*	Expression does not change	Expression does not change	No significant impact on prognosis
*CYP4F3*	Expression does not change	N/A	No significant impact on prognosis
*CYP4X1*	Lower expression in the tumor	Lower expression in the tumor	No significant impact on prognosis
*GPR75*	Expression does not change	Expression does not change	No significant impact on prognosis

Red background—higher expression in the tumor; blue background—lower expression in the tumor; red background—worse prognosis with higher expression.

**Table 10 cancers-15-00946-t010:** Pan-cancer analysis of the expression of the cytochromes P450 and GPR75 receptor genes in question.

Name of Cancer	*CYP1A2*	*CYP1B1*	*CYP2C8*	*CYP2C9*	*CYP2C19*	*CYP2J2*	*CYP2U1*	*CYP3A4*	*CYP4A11*	*CYP4F2*	*CYP4F3*	*CYP4X1*	*GPR75*
Adrenocortical carcinoma (ACC)	=	↓	↓	=	=	=	=	=	=	=	=	=	=
Bladder urothelial carcinoma (BLCA)	=	↓	=	=	=	=	↓	=	=	=	=	=	=
Breast invasive carcinoma (BRCA)	=	=	=	=	=	=	=	=	=	=	=	=	=
Cervical squamous cell carcinoma and endocervical adenocarcinoma (CESC)	=	↓	=	=	=	↑	↓	=	=	=	↑	=	=
Cholangiocarcinoma (CHOL)	↓	↑	↓	↓	↓	↓	=	↓	↓	↓	↓	=	=
Colon adenocarcinoma (COAD)	=	↓	=	=	=	↑	=	=	=	↑	↑	=	=
Lymphoid neoplasm diffuse large B-cell lymphoma (DLBC)	=	=	=	=	=	=	↑	=	=	=	↓	=	=
Esophageal carcinoma (ESCA)	=	=	=	↓	=	↓	=	=	=	=	↓	↓	=
Glioblastoma multiforme (GBM)	=	=	↓	=	=	=	↑	=	=	=	=	↓	=
Head and neck squamous cell carcinoma (HNSC)	=	=	=	=	=	↓	=	=	=	=	=	↓	=
Kidney chromophobe (KICH)	=	↓	=	=	=	=	=	=	↓	↓	↓	↓	=
Kidney renal clear cell carcinoma (KIRC)	=	↓	=	=	=	↑	=	=	↓	↓	↓	=	=
Kidney renal papillary cell carcinoma (KIRP)	=	↓	=	=	=	=	=	=	↓	↓	↓	↓	=
Acute myeloid leukemia (LAML)	=	↑	=	=	=	=	↑	=	=	=	=	=	↑
Brain lower grade glioma (LGG)	=	=	↓	=	=	=	↑	=	=	=	=	↓	↑
Liver hepatocellular carcinoma (LIHC)	↓	=	↓	↓	↓	=	=	↓	↓	=	=	=	=
Lung adenocarcinoma (LUAD)	=	=	=	=	=	↑	=	=	=	=	=	=	=
Lung squamous cell carcinoma (LUSC)	=	=	=	=	=	=	=	=	=	=	=	=	=
Ovarian serous cystadenocarcinoma (OV)	=	=	=	=	=	↑	=	=	=	=	=	↑	=
Pancreatic adenocarcinoma (PAAD)	=	↑	=	↑	=	↑	↑	↓	=	=	↑	=	=
Pheochromocytoma and paraganglioma (PCPG)	=	=	=	=	=	=	↑	↓	=	=	=	=	=
Prostate adenocarcinoma (PRAD)	=	=	=	=	=	↑	=	=	=	=	=	=	=
Rectum adenocarcinoma (READ)	=	↓	=	=	=	↑	=	=	=	↑	↑	=	=
Sarcoma (SARC)	=	=	=	=	=	=	=	=	↓	↓	↓	=	=
Skin cutaneous melanoma (SKCM)	=	=	=	=	=	↓	↑	↓	=	=	↓	↓	=
Stomach adenocarcinoma (STAD)	=	=	↓	↓	=	↑	=	=	=	=	↑	↓	=
Testicular germ cell tumors (TGCT)	=	=	↓	=	=	↓	=	=	=	↓	=	=	=
Thyroid carcinoma (THCA)	=	↑	=	=	=	=	=	=	=	=	=	↓	=
Thymoma (THYM)	=	↑	=	=	=	=	↑	=	=	=	↓	=	=
Uterine corpus endometrial carcinoma (UCEC)	=	↓	=	=	=	↑	↓	=	=	=	=	=	=
Uterine carcinosarcoma (UCS)	=	↓	=	=	=	↑	↓	=	=	=	=	=	=

Red background, ↑—expression higher in tumor than in healthy tissue; blue background, ↓—expression lower in tumor than in healthy tissue; gray background, =—expression does not differ between tumor and healthy tissue.

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
