# Peer review of "Synthesis and Significance of Arachidonic Acid, a Substrate for Cyclooxygenases, Lipoxygenases, and Cytochrome P450 Pathways in the Tumorigenesis of Glioblastoma Multiforme, Including a Pan-Cancer Comparative Analysis"

_cancers, 2023, doi:10.3390/cancers15030946_

Round 1
Reviewer 1 Report
The authors made a quality and detail overview of literature data regarding arachidonic acid metabolism in tumorigenesis of glioblastoma multiforme.
In this review the authors give detail descriptions of arachidonic acid metabolic pathway in general in a section and overview of literature data regarding the same metabolic pathway in tumorigenesis of glioblastoma multiforme in the next one section, so the structure of the article is well defined. At the same time, all biologically significant metabolic pathways are described in the manuscript.
The manuscript may be accepted in current form, and I only send some optional suggestions that may or may not be accepted.
The search of literature is well described by the authors could add some key word that they used, whether they included only data for clinical trials or also from review articles, did they consider animal studies (if there is any) or only human studies.
Besides, I have a minor suggestion that the authors change the term our results (line 133) and our study did not show (line 138) with a maybe more appropriate term previous results of our research group.
Author Response
Rev.1.
This review focuses on the role of ARA and its metabolism in glioblastoma. Two bioinformatic sources are the backbone of the paper. Transcriptomic analyses are extended with information from articles. The review is detailed, yet well structured.
Comments:
1. I believe that the abstract should include a reference to the methodological approach.
The abstract has been changed according to the recommendations.
Chapters 3, 5 and 6 contain detailed diagrams and summary tables (only in Chapters 5 and 6). Similar diagrams and tables should also have been in Chapters 4 and 7.
Figures and tables have been added to Chapters 4 and 7.
- The description of the columns in the right part of Tables 1 and 2 is not very clear. Instead of a redundant description below the table, I would like to see at least an abbreviated description directly in the table.
The tables have been modified according to the recommendations.
- In Chapter 7 The authors mention that ARA C20:4n-6 is converted by CYP4A family but they further discuss only two members of this family. What about the other members? See, e.g. PMID: 18591218. Overall, Chapter 7 is very brief, especially compared to the other chapters.
The article is quoted in the paper. The PMID: 18591218 study examined CYP4A1 in a rat model. In current nomenclature, CYP4A1 is a rat enzyme, not human. For this reason, we describe the possible associations of human CYP4A11 in GBM. The subsection on cytochrome p450 pathway has been significantly extended.
The manuscript has been proofread by a native speaker. Certificate attached.

Reviewer 2 Report
IIn the article „Synthesis and Significance of Arachidonic Acid, a Substrate for Cyclooxygenases and Lipoxygenases Pathways, in the Tumorigenesis of Glioblastoma Multiforme ", authors retrieved information [Pubmed, Seifert et al (ref. 8), GEPIA database (ref. 9) and Rembrandt datasets] about Arachidonic Acid (ARA) derived lipid mediators/components to discuss tumorigenic pathways in GBM. To support their concept, authors also referred to some previously published articles where these components were randomly tested/predicted/discussed. I have several concerns about this study; some of major ones are below:
- In multiple instances, the authors did not agree with the results from the literature especially when comparing their own actual results and most contradictions arose when their two used datasets (Seifert et al. and GEPIA database) produced conflicting outcome. This issue remained unresolved throughout the manuscript.
- Another problem I see is- most of the sections only explain the biosynthesis pathways along with the by-products and very little about the GBM-related data. In my opinion, the authors should have directly displayed the arrows (up/down) in the Figures 1-5, so that readers could get a clear picture of the GBM-related changes in these components. In addition, most importantly, by using two different colors to indicate whether two datasets (GEPIA and Seifert et al.) have similar results or only one of them.
- Undoubtedly, some of these components have also been observed to be up- or down-regulated in other cancers. A brief description about them, linking with the current results (Table 1 &2) would be necessary to rule out the possibility that their dysregulation is GBM-specific or a normal occurrence in the cancers.
- In my opinion, it is necessary to discuss the outlook for key components (COX-2, mPGES-1, AKR1B1, GGT5), whose higher expression was correlated with a worse prognosis in this study.
Minor comment-
- Authors wrote- (Line 1120-1121-We are talking, for example, about EET, lipoxins, hepoxilins, and some prostanoids including PGF2α and TxA2. Investigating the function of these compounds will provide a better understanding of GBM tumor function). This line is unclear, considering the very short half-life of some of these components, which investigations authors are pointed to.
- There is mistake in Line 534 (Similarly, Seifert et al. showed that in a GBM..) This reference is not Seifert et al.
In my opinion, the overall concept is interesting; however, it is not properly executed. Besides, there is no clear conclusion due to the discrepancy between the used datasets (GEPIA and Seifert et al.). The outlook certainly is not satisfactory.
Author Response
Rev.2.
In the article „Synthesis and Significance of Arachidonic Acid, a Substrate for Cyclooxygenases and Lipoxygenases Pathways, in the Tumorigenesis of Glioblastoma Multiforme ", authors retrieved information [Pubmed, Seifert et al (ref. 8), GEPIA database (ref. 9) and Rembrandt datasets] about Arachidonic Acid (ARA) derived lipid mediators/components to discuss tumorigenic pathways in GBM. To support their concept, authors also referred to some previously published articles where these components were randomly tested/predicted/discussed. I have several concerns about this study; some of major ones are below:
- In multiple instances, the authors did not agree with the results from the literature especially when comparing their own actual results and most contradictions arose when their two used datasets (Seifert et al. and GEPIA database) produced conflicting outcome. This issue remained unresolved throughout the manuscript.
Clarification has been added. This was most likely the result of studying different populations.
- Another problem I see is- most of the sections only explain the biosynthesis pathways along with the by-products and very little about the GBM-related data. In my opinion, the authors should have directly displayed the arrows (up/down) in the Figures 1-5, so that readers could get a clear picture of the GBM-related changes in these components. In addition, most importantly, by using two different colors to indicate whether two datasets (GEPIA and Seifert et al.) have similar results or only one of them.
In our manuscript, based on available scientific information (PubMed), we described the importance of ARA-derived lipids mediators on tumor mechanisms in GBM. At the beginning of each chapter, we described the synthesis pathways. This is kind of like a theoretical introduction in which we did not discuss the importance of the lipid mediators in GBM, but only general information. It is meant to prepare the reader for the subsections where we discuss the effects on GBM.
The figures have been modified.
- Undoubtedly, some of these components have also been observed to be up- or down-regulated in other cancers. A brief description about them, linking with the current results (Table 1 &2) would be necessary to rule out the possibility that their dysregulation is GBM-specific or a normal occurrence in the cancers.
A pan-cancer analysis of the expression of all the genes described in this paper was performed. In these analyses, GBM was compared with other cancers and also with lower-grade gliomas.
- In my opinion, it is necessary to discuss the outlook for key components (COX-2, mPGES-1, AKR1B1, GGT5), whose higher expression was correlated with a worse prognosis in this study.
A relevant passage has been added to the article.
Minor comment-
- Authors wrote- (Line 1120-1121-We are talking, for example, about EET, lipoxins, hepoxilins, and some prostanoids including PGF2α and TxA2. Investigating the function of these compounds will provide a better understanding of GBM tumor function). This line is unclear, considering the very short half-life of some of these components, which investigations authors are pointed to.
Sentences have been added to clarify this thought. Unstable lipid mediators can affect tumor processes only near the site of synthesis.
- There is mistake in Line 534 (Similarly, Seifert et al. showed that in a GBM..) This reference is not Seifert et al.
It has been corrected. The sentence refers to GEPIA.
In my opinion, the overall concept is interesting; however, it is not properly executed. Besides, there is no clear conclusion due to the discrepancy between the used datasets (GEPIA and Seifert et al.). The outlook certainly is not satisfactory.
The manuscript has been proofread by a native speaker. Certificate attached.

Reviewer 3 Report
This review focuses on the role of ARA and its metabolism in glioblastoma. Two bioinformatic sources are the backbone of the paper. Transcriptomic analyses are extended with information from articles. The review is detailed, yet well structured.
Comments:
1. I believe that the abstract should include a reference to the methodological approach.
2. Chapters 3, 5 and 6 contain detailed diagrams and summary tables (only in Chapters 5 and 6). Similar diagrams and tables should also have been in Chapters 4 and 7.
3. The description of the columns in the right part of Tables 1 and 2 is not very clear. Instead of a redundant description below the table, I would like to see at least an abbreviated description directly in the table.
4. In Chapter 7 The authors mention that ARA C20:4n-6 is converted by CYP4A family but they further discuss only two members of this family. What about the other members? See, e.g. PMID: 18591218. Overall, Chapter 7 is very brief, especially compared to the other chapters.
Author Response
Rev.3.
The authors made a quality and detail overview of literature data regarding arachidonic acid metabolism in tumorigenesis of glioblastoma multiforme.
In this review the authors give detail descriptions of arachidonic acid metabolic pathway in general in a section and overview of literature data regarding the same metabolic pathway in tumorigenesis of glioblastoma multiforme in the next one section, so the structure of the article is well defined. At the same time, all biologically significant metabolic pathways are described in the manuscript.
The manuscript may be accepted in current form, and I only send some optional suggestions that may or may not be accepted.
The search of literature is well described by the authors could add some key word that they used, whether they included only data for clinical trials or also from review articles, did they consider animal studies (if there is any) or only human studies.
In our work, we tried not to use review papers. They often cite previous review papers that cite other review papers. Because of this, it is very difficult to get to the actual experimental papers. Therefore, we tried to use only experimental works, especially on human models.
Besides, I have a minor suggestion that the authors change the term our results (line 133) and our study did not show (line 138) with a maybe more appropriate term previous results of our research group.
In accordance with the reviewer's guidelines, this has been modified.
The manuscript has been proofread by a native speaker. Certificate attached.
